# TIME-EFFICIENT REINFORCEMENT LEARNING WITH STOCHASTIC STATEFUL POLICIES

**Firas Al-Hafez**[1]**, Guoping Zhao**[2]**, Jan Peters**[1,3]**, Davide Tateo**[1]

[1] Intelligent Autonomous Systems, [2] Locomotion Laboratory
[3] German Research Center for AI (DFKI), Centre for Cognitive Science, Hessian.AI
TU Darmstadt, Germany
{name.surname}@tu-darmstadt.de

## ABSTRACT

Stateful policies play an important role in reinforcement learning, such as handling partially observable environments, enhancing robustness, or imposing an inductive bias directly into the policy structure. The conventional method for training stateful policies is Backpropagation Through Time (BPTT), which comes with significant drawbacks, such as slow training due to sequential gradient propagation and the occurrence of vanishing or exploding gradients. The gradient is often truncated to address these issues, resulting in a biased policy update. We present a novel approach for training stateful policies by decomposing the latter into a stochastic internal state kernel and a stateless policy, jointly optimized by following the *stateful policy gradient*. We introduce different versions of the stateful policy gradient theorem, enabling us to easily instantiate stateful variants of popular reinforcement learning and imitation learning algorithms. Furthermore, we provide a theoretical analysis of our new gradient estimator and compare it with BPTT. We evaluate our approach on complex continuous control tasks, e.g. humanoid locomotion, and demonstrate that our gradient estimator scales effectively with task complexity while offering a faster and simpler alternative to BPTT.[1]

## 1 INTRODUCTION

Stateful policies are a fundamental tool for solving complex Reinforcement Learning (RL) problems. These policies are particularly relevant for RL in a Partially Observable Markov Decision Process (POMDP), where the history of interactions needs to be processed at each time step. Stateful policies, such as a Recurrent Neural Network (RNN), compress the history into a latent recurrent representation, allowing them to deal with the ambiguity of environment observations. Ni et al. (2022) have shown that RNN policies can be competitive and even outperform specialized algorithms in many POMDP tasks. Besides POMDPs, stateful policies can be used to incorporate inductive biases directly into the policy, e.g. a stateful oscillator to learn locomotion Ijspeert et al. (2007); Bellegarda & Ijspeert (2022a), or to solve Meta-RL tasks by observing the history of rewards Ni et al. (2022).

Existing methods for stateful policy learning mostly rely either on black-box/evolutionary optimizers or on the BPTT algorithm. While black-box approaches struggle with high-dimensional parameter spaces, making them less suitable for neural approximators, BPTT has shown considerable success in this domain Bakker (2001); Wierstra et al. (2010); Meng et al. (2021). However, BPTT suffers from significant drawbacks, including the need for sequential gradient propagation through trajectories, which significantly slows down training time and can lead to vanishing or exploding gradients. To address these issues, practical implementations often use a truncated history, introducing a bias into the policy update and limiting memory to the truncation length. Furthermore, integrating BPTT into standard RL algorithms is not straightforward due to the requirement of handling sequences of varying lengths instead of single states.

In this paper, we propose an alternative solution to compute the gradient of a stateful policy. We decompose the stateful policy into a stochastic policy state kernel and a conventional stateless policy, which are jointly optimized by following the Stochastic Stateful Policy Gradient (S2PG). By

---

[1]The code is available at: https://github.com/robfiras/s2pg

adopting this approach, we can train arbitrary policies with an internal state without BPTT. S2PG not only provides an unbiased gradient update but also accelerates the training process and avoids issues like vanishing or exploding gradients. In addition, our approach can be applied to any existing RL algorithm by modifying a few lines of code. We also show how to extend the S2PG theory to all modern RL approaches by introducing different versions of the policy gradient theorem for stateful policies. This facilitates the instantiation of stateful variations of popular RL algorithms like Soft-Actor Critic (SAC) Haarnoja et al. (2018), Twin Delayed DDPG (TD3) Fujimoto et al. (2018), Proximal Policy Optimization (PPO) Schulman et al. (2017), as well as Imitation Learning (IL) algorithms like Generative Adversarial Imitation Learning (GAIL) Ho & Ermon (2016) and Least-Squares Inverse Q-Learning (LS-IQ) Al-Hafez et al. (2023). Our approach is also applicable to settings where the critic has only access to observations. In such cases, we combine our method with Monte-Carlo rollouts, as typically done in PPO. In the off-policy scenario, we use a critic with privileged information Pinto et al. (2018); Lee et al. (2020b); Peng et al. (2018).

To evaluate our approach, we conduct a theoretical analysis on the variance of S2PG and BPTT, introducing two new bounds and highlighting the behavior of each estimator in different regimes. Empirically, we compare the performances in common continuous control tasks within POMDPs when using RL. To illustrate the potential use of S2PG for inductive biases in policies, we give an example of a trainable Ordinary Differential Equation (ODE) of an oscillator used in the policy together with an Feed-Forward Network (FFN) to achieve walking policies under complete blindness. Finally, We also demonstrate the scalability of our method by introducing two complex locomotion tasks under partial observability and dynamics randomization. Our results indicate that these challenging tasks can be effectively solved using stateful IL algorithms, and demonstrate that S2PG offers a simple and efficient alternative to BPTT that scales well with task complexity.

**Related Work.** RNNs are the most popular type of stateful policy. Among RNNs, gated recurrent networks, such as Long-Short Term Memory (LSTM) Hochreiter & Schmidhuber (1997) or Gated Recurrent Units (GRU) Cho et al. (2014) networks, are the most prominent ones Ni et al. (2022); Wierstra et al. (2010); Meng et al. (2021); Heess et al. (2015); Espeholt et al. (2018); Yang & Nguyen (2021). Wierstra et al. (2010) introduced the *recurrent policy gradient* using LSTMs and the GPOMDP Baxter & Bartlett (2001) algorithm, where BPTT was used to train the trajectory. Many subsequent works have adapted BPTT-based training of RNNs to different RL algorithms. Heess et al. (2015) introduced the actor-critic Recurrent Deterministic Policy Gradient (RDPG) and Recurrent Stochastic Value Gradient (RSVG), which were later updated to the current state-of-the-art algorithms like Recurrent Twin-Delayed Deep Deterministic Policy Gradient (RTD3) and Recurrent Soft Actor-Critic (RSAC) Yang & Nguyen (2021). Many works utilize separate RNNs for the actor and the critic in RTD3 and RSAC, as it yields significant performance gains compared to a shared RNN Ni et al. (2022); Meng et al. (2021); Heess et al. (2015); Yang & Nguyen (2021). Although LSTMs have a more complex structure, GRUs have demonstrated slightly better performance in continuous control tasks Ni et al. (2022). Stateful Policies can also be used to encode an inductive bias into the policy. For instance, a Central Pattern Generator (CPG) is a popular choice for locomotion tasks Ijspeert (2008); Bellegarda & Ijspeert (2022b); Campanaro et al. (2021). We conduct experiments with CPGs in Section E.2, where we also present relevant related work. Using a similar approach to the one presented in this paper, Zhang et al. (2016) introduce a stochastic internal state transition kernel. In this approach, states are considered as a memory that the policy can read and write. However, they did not extend their approach to the actor-critic case, which is the main focus of our work. Rakelly et al. (2019) proposed a Meta-RL approach to learn a distribution over latent variables that encodes history to extract context variables. Similarily, other approaches such as DVRL Igl et al. (2018) and SLAC Lee et al. (2020a) force the recurrent state to be a belief state, exploiting a learned environment model. An alternative to stateful policies, particularly useful in POMDPs settings, is to use a history of observations. These policies use either fully connected MLPs, time convolution Lee et al. (2020b), or transformers Lee et al. (2023); Radosavovic et al. (2023). It can be shown that for some class of Markov Decision Processs (MDPs), these architectures allow learning near-optimal policies Efroni et al. (2022). The main drawback is that they require high dimensional input, store many transitions in the buffer, and cannot encode inductive biases in the latent space. Finally, the setting where privileged information from the simulation is used is widespread in sim-to-real robot learning Lee et al. (2020b); Peng et al. (2018). Compared to so-called teacher-student approaches Lee et al. (2020b), which learn a privileged policy and then train a recurrent policy using behavioral cloning, our approach can learn a recurrent policy online using privileged information for the critic.

## 2   STOCHASTIC STATEFUL POLICY GRADIENTS

**Preliminaries.**   An MDP is a tuple $(\mathcal{S}, \mathcal{A}, P, r, \gamma, \iota)$, where $\mathcal{S}$ is the state space, $\mathcal{A}$ is the action space, $P : \mathcal{S} \times \mathcal{A} \times \mathcal{S} \to \mathbb{R}^+$ is the transition kernel, $r : \mathcal{S} \times \mathcal{A} \to \mathbb{R}$ is the reward function, $\gamma$ is the discount factor, and $\iota : \mathcal{S} \to \mathbb{R}^+$ is the initial state distribution. At each step, the agent observes a state $s \in \mathcal{S}$ from the environment, predicts an action $a \in \mathcal{A}$ using the policy $\pi : \mathcal{S} \times \mathcal{A} \to \mathbb{R}^+$, and transitions with probability $P(s'|s, a)$ into the next state $s' \in \mathcal{S}$, where it receives the reward $r(s, a)$. We define an occupancy measure $\rho^{\pi_\theta}(s, z) = \sum_{t=0}^{\infty} \gamma^t Pr\{s = s_t \wedge z = z_t | \pi_\theta\}$ giving metric on how frequently the tuple $(s, z)$ is visited, where $z_t$ is the internal state of the policy. The POMDP is tuple $(\mathcal{S}, \mathcal{A}, \mathcal{O}, P, O, r, \gamma, \iota)$ extending the MDP by the observation space $\mathcal{O}$ and the conditional probability density $O : \mathcal{O} \times \mathcal{S} \to \mathbb{R}^+$, such that for every state $s \in \mathcal{S}$, the probability density of $o \in \mathcal{O}$ is $O(o|s)$. A trajectory $\tau$ is a (possibly infinite) sequence of states, actions, and observations sampled by interacting with the environment under a policy $\pi$. The return of a trajectory is defined as $J(\tau) = \sum_{i=0}^{\infty} \gamma^t r(s_t, a_t)$. We define the observed history $h_t = \langle s_0, \ldots s_t, \rangle$ as the sequence of states taken up to the current timestep $t$. In the partially observable case, the history is given by observations and actions such that $h_t = \langle o_0, a_0 \ldots o_{t-1}, a_{t-1}, o_t \rangle$. In general, an optimal policy for a POMDP depends on the full observed history $h_t$, i.e. $a \sim \pi(\cdot|h_t)$. However, most of the time, we can substitute $h_t$ with sufficient statistics $z_t = S(h_t)$, namely the internal state of the policy.

### 2.1   POLICY GRADIENT OF STATEFUL POLICIES

Wierstra et al. (2010) presented the first implementation of the policy gradient for recurrent policies. This formulation is based on the likelihood ratio trick, i.e. the score function estimator. To obtain this formulation, the authors consider a generic history-dependant policy $\nu(a|h_t)$, obtaining the following gradient formulation

$$\nabla_{\boldsymbol{\theta}} \mathcal{J}(\nu_{\boldsymbol{\theta}}) = \mathbb{E}_{\tau} \left[ \sum_{t=0}^{T-1} \nabla_{\boldsymbol{\theta}} \log \nu_{\boldsymbol{\theta}}(a_t|h_t) J(\tau) \right]. \tag{1}$$

Note that each $\nabla_{\boldsymbol{\theta}} \log \nu_{\boldsymbol{\theta}}(a_t|h_t)$ term in equation 1 depends on the whole history $h_t$ at each timestep $t$ which makes BPTT necessary for stateful policies, such as RNNs. However, BPTT comes with many issues. Firstly, this method is inherently sequential, limiting the utilization of potential gradient computations that could be performed in parallel. Secondly, when dealing with long trajectories, the gradients may suffer from the problems of exploding or vanishing gradients, which can hinder the learning process. Therefore, histories are often truncated, leading to a *biased* gradient estimate.

To solve these issues, we consider a different policy structure. Instead of looking at policies with an internal state, we model our policy as a joint probability distribution over actions and next internal states, i.e. $(a, z') \sim \pi_\theta(\cdot|s, z)$. By considering a stochastic policy state transition, we derive a policy gradient formulation that does not require the propagation of the policy gradient through time but only depends on the local information available, i.e. the (extended) state transition.

**Lemma 2.1.** *(Stochastic Stateful Policy Gradient) Let $\pi_{\boldsymbol{\theta}}(a, z'|s, z)$ be a parametric policy representing the joint probability density used to generate the action $a$ and the next internal state $z'$, and let $\bar{\tau}$ be the extended trajectory including the internal states. Then, the S2PG can be written as*

$$\nabla_{\boldsymbol{\theta}} \mathcal{J}(\pi_{\boldsymbol{\theta}}) = \mathbb{E}_{\bar{\tau}} \left[ \sum_{t=0}^{T-1} \nabla_{\boldsymbol{\theta}} \log \pi_{\boldsymbol{\theta}}(a_t, z_{t+1}|s_t, z_t) J(\bar{\tau}) \right]. \tag{2}$$

This lemma is straightforward to derive using the classical derivation of the policy gradient. The complete proof can be found in Appendix A.1. In contrast to the gradient in equation 1, the gradient in equation 2 depends on the information of the *current* timestep $t$, i.e. $a_t$, $z_{t+1}$, $s_t$ and $z_t$. We can deduce the following from Lemma 2.1:

**The gradient of a stateful policy can be computed stochastically**, by treating the internal state of policy as a random variable. This modification results in an algorithm that does not require BPTT, but only access to samples from $z$.

**We allow to trade off speed and accuracy while keeping the gradient estimation *unbiased*.** While truncating the history to save computation time biases the gradient, estimating the expectation with a finite amount of samples of $z$ does not. In contrast to BPTT, this formulation supports parallel computation of the gradient, allowing to fully exploit the parallelization capabilities of modern

Figure 1: Stochastic computational graph illustrating, left to right, a comparison of a stateless policy, a stateful policy trained with BPTT, and a stateful policy incorporating a stochastic internal state transition kernel. Deterministic nodes (squares) allow for the passage of analytic gradients, while stochastic nodes (circles) interrupt the deterministic paths. The input node to the graph is represented by $\theta$. The blue lines indicate the deterministic paths for which analytic gradients are available for a specific action. For a detailed demonstration of how stochastic node gradients are calculated compared to deterministic ones, refer to Figure 5 in Appendix B

automatic differentiation tools and parallel simulation environments. As the batch size for gradient computation increases, the variance of our gradient estimator decreases, as shown later in Section 3.

We present an alternative perspective by explaining the core principle of Lemma 2.1 in the context of stochastic computation graphs, as proposed by Schulman et al. (2015a). The key idea of Lemma 2.1 is to shield the deterministic path through internal states using a stochastic node, specifically our stochastic internal state kernel. Figure 1 illustrates this by comparing analytic gradient paths of the internal state $z$ across different policies. While S2PG resolves issues with exploding and vanishing gradients in long sequences, it introduces potential high variance due to the stochastic internal state kernel. To address this, we propose compensating for the added variance by utilizing state-of-the-art actor-critic methods while maintaining the same policy structure.

## 2.2 ACTOR-CRITIC METHODS FOR STATEFUL POLICIES

To define actor-critic methods, we need to extend the definitions of value functions into the stateful policy setting. Let $s$ be some state of the environment, $z$ some state of the policy, $a$ a given action, and $z'$ the next policy state. We define the state-action value function of our policy $\pi(a, z'|s, z)$ as

$$Q^\pi(s, z, a, z') = r(s, a) + \gamma \mathop{\mathbb{E}}_{s'} \left[ \mathop{\mathbb{E}}_{a', z''} \left[ Q^\pi(s', z', a', z'') \right] \right] = r(s, a) + \gamma \mathop{\mathbb{E}}_{s'} \left[ V^\pi(s', z') \right] \qquad (3)$$

with $s' \sim P(\cdot|s, a)$ and $(a', z'') \sim \pi(\cdot|s', z')$. Furthermore, we define the state-value function as

$$V^\pi(s, z) = \mathop{\mathbb{E}}_{a, z'} \left[ Q^\pi(s, z, a, z') \right] \qquad \text{where } (a, z') \sim \pi(\cdot|s, z). \qquad (4)$$

Note that these definitions are equivalent to those of the value and action-value functions in a Markov policy. Indeed, the value function $V(s, z)$ is the expected discounted return achieved by the policy $\pi$ starting from the state $s$ with initial internal state $z$. Thus, we can write the expected discounted return of a stateful policy as $\mathcal{J}(\pi) = \int \iota(s, z) V(s, z) ds dz$ with the initial state distribution $\iota(s, z)$.

The fundamental theorem of actor-critic algorithms is the Policy Gradient Theorem (PGT) Sutton et al. (1999). This theorem lays the connection between value functions and policy gradients. Many successful practical approaches are based on this idea and constitute a fundamental part of modern RL. Using our stochastic internal state kernel and the definitions of value and action-value function in equation 4 and 3, we can provide a straightforward derivation of the PGT for stateful policies.

**Theorem 2.2.** *(Stateful Policy Gradient Theorem) Let $\rho^{\pi_\theta}(s, z)$ be the occupancy measure and $Q^{\pi_\theta}(s, z, a, z')$ be the value function of a parametric policy $\pi_\theta(a, z'|s, z)$, then*

$$\nabla_\theta \mathcal{J}_\theta = \int_\mathcal{S} \int_\mathcal{Z} \rho^{\pi_\theta}(s, z) \int_\mathcal{A} \int_\mathcal{Z} \nabla_\theta \pi_\theta(a, z'|s, z) Q^{\pi_\theta}(s, z, a, z') \, dz' \, da \, dz \, ds,$$

*is the gradient of the stateful policy.*

The proof follows the lines of the original PGT and is given in Appendix A.3. Differently from Sutton et al. (1999), which considers only the discrete action setting, we provide the full derivation for continuous state and action spaces.

Theorem 2.2 elucidates the key idea of this work:

**We learn a $Q$-function to capture the values of internal state transitions**, compensating the additional variance in the policy gradient estimate induced by our stochastic internal state transition kernel. This approach enables the efficient training of stateful policies without the need for BPTT, effectively reducing the computational time required to train stateful policies to that of stateless policies. Figure 5 in Appendix B further illustrates and reinforces this concept.

While the PGT is a fundamental building block of actor-critic methods, some of the more successful approaches, namely the Trust Region Policy Optimization (TRPO) Schulman et al. (2015b) and the PPO Schulman et al. (2017) algorithm, are based on the Performance Difference Lemma Kakade & Langford (2002). We can leverage the simplicity of our policy structure to derive Lemma A.1, a modified version of the original Lemma for stateful policies, allowing us to implement recurrent versions of PPO and TRPO with S2PG. The Lemma and the proof can be found in Appendix A.4.

## 2.3 THE DETERMINISTIC LIMIT

The last important piece of policy gradient theory is the definition of the Deterministic Policy Gradient Theorem (DPGT) Silver et al. (2014). This theorem can be seen as the deterministic limit of the PGT for some class of probability distributions. DPGT provides the basis for many successful actor-critic algorithms, such as Deep Deterministic Policy Gradient (DDPG) Lillicrap et al. (2016) and TD3 Fujimoto et al. (2018). We can derive the DPGT for our stateful policies:

**Theorem 2.3.** *(Stateful Deterministic Policy Gradient Theorem)* Let $\mu_{\boldsymbol{\theta}}(s,z) = \begin{bmatrix} \mu_{s,z,\boldsymbol{\theta}}^a, \mu_{s,z,\boldsymbol{\theta}}^z \end{bmatrix}^\top$ *be a deterministic policy, where $\mu_{s,z,\boldsymbol{\theta}}^a = \mu_{\boldsymbol{\theta}}^a(s,z)$ represent the action model and $\mu_{s,z,\boldsymbol{\theta}}^z = \mu_{\boldsymbol{\theta}}^z(s,z)$ represents the internal state transition model. Then, under mild regularity assumptions, the policy gradient of the stateful deterministic policy under $\rho_{s,z}^{\mu_{\boldsymbol{\theta}}} = \rho^{\mu_{\boldsymbol{\theta}}}(s,z)$ is*

$$\nabla_{\boldsymbol{\theta}}\mathcal{J}(\mu_{\boldsymbol{\theta}}) = \int_{\mathcal{S},\mathcal{Z}} \rho_{s,z}^{\mu_{\boldsymbol{\theta}}} \left( \nabla_{\boldsymbol{\theta}}\mu_{s,z}^a \nabla_a Q^{\mu_{\boldsymbol{\theta}}}(s,z,a,\mu_{s,z}^z)\big|_{a=\mu_{s,z}^a} + \nabla_{\boldsymbol{\theta}}\mu_{s,z}^z \nabla_{z'} Q^{\mu_{\boldsymbol{\theta}}}(s,z,\mu_{s,z}^a,z')\big|_{z'=\mu_{s,z}^z} \right) ds\,dz.$$

The proof is given in Appendix A.5. It is important to notice that, while the deterministic policy gradient theorem allows computing the gradient of a deterministic policy, it assumes the knowledge of the $Q$-function for the deterministic policy $\mu_{\boldsymbol{\theta}}$. However, to compute the $Q$-function of a given policy it is necessary to perform exploration. Indeed, to accurately estimate the $Q$-values, actions different from the deterministic policy needs to be taken. We want to stress that in this scenario, instead of using the standard $Q$-function formulation, we use our definition of $Q$-functions for stateful policies: therefore, we must also explore the internal state transition. This key concept explains why in our formulation it is not necessary to perform BPTT even in the deterministic policy scenario.

## 2.4 THE PARTIALLY OBSERVABLE SETTING

Within this section, we introduce the POMDP settings used within this work and adapt the previous theory accordingly. Therefore, we extend Lemma 2.1 as follows.

**Corollary 2.4.** *The stateful policy gradient in partially observable environments is*

$$\nabla_{\boldsymbol{\theta}}\mathcal{J}(\pi_{\boldsymbol{\theta}}) = \mathbb{E}_{\substack{\bar{\tau} \\ o_t \sim O(s_t)}} \left[ \sum_{t=0}^{T-1} \nabla_{\boldsymbol{\theta}} \log \pi_{\boldsymbol{\theta}}(a_t, z_{t+1}|o_t, z_t) J(\bar{\tau}) \right].$$

The proof is given in Appendix A.2. Corollary 2.4 offers a particularly intriguing perspective: If we have knowledge of $J(\bar{\tau})$, it is possible to learn an internal state representation $z_t$ that effectively encapsulates past information *without* looking back in time. In other words, it enables to learn the compression of past information without explicitly considering previous time steps. This stands in stark contrast to BPTT, where all past information is used to learn a belief of the current state. Similarly to Corollary 2.4, we can extend Theorem 2.2

$$\nabla_{\boldsymbol{\theta}}\mathcal{J}_{\boldsymbol{\theta}} = \int_{\mathcal{S}} \int_{\mathcal{O}} \int_{\mathcal{Z}} \rho^{\pi_{\boldsymbol{\theta}}}(s,z)\, O(o|s) \int_{\mathcal{A}} \int_{\mathcal{Z}} \nabla_{\boldsymbol{\theta}}\pi_{\boldsymbol{\theta}}(a,z'|o,z)\, Q^{\pi_{\boldsymbol{\theta}}}(s,z,a,z')\, da\,ds\,. \tag{5}$$

It is important to note the distinction between the $Q$-function, which receives states, and the policy, which relies on observations only. While it is possible to learn a $Q$-function on observations

via bootstrapping, the task of temporal credit assignment in POMDPs is very challenging, often requiring specialized approaches Igl et al. (2018); Lee et al. (2020a). This difficulty stems from the following causality dilemma: On one hand, the policy can learn a condensed representation of the history assuming accurate $Q$-value estimates. On the other hand, the $Q$-function can be learned using bootstrapping techniques assuming a reliable condensed representation of the history. Because both, bootstrapping a $Q$-value function and learning a representation of the history, rely on meaningful representations of the other, doing both at the same time often results in unstable training. We show in Section 4 that it is possible to combine Monte-Carlo estimates with a critic that uses observations only using PPO to learn in a true POMDP setting. However, we limit ourselves to the setting with privileged information in the critic, as shown in Equation 5, for approaches that learn a value function via bootstrapping.

## 3 VARIANCE ANALYSIS OF THE GRADIENT ESTIMATORS

To theoretically assess our estimator's variance, we provide the upper bounds of BPTT and S2PG in the Gaussian policy setting with constant covariance matrix $\Sigma$. As done in previous work Papini et al. (2022); Zhao et al. (2011), we define the variance of a random vector as the trace of its covariance matrix (see equation equation 17 in Appendix A.6). Furthermore, for both gradient estimators, we assume that the function approximator for the mean has the following structure

$$\mu_{\boldsymbol{\theta}}^a(s_t, z_t) = f_{\boldsymbol{\theta}}(s_t, z_t) \quad \mu_{\boldsymbol{\theta}}^{\tilde{z}}(s_t, z_t) = \eta_{\boldsymbol{\theta}}(s_t, z_t) \text{ (for S2PG)} \quad z_{t+1} = \eta_{\boldsymbol{\theta}}(s_t, z_t) \text{ (for BPTT)}. \tag{6}$$

Hence, $f_{\boldsymbol{\theta}}(s_t, z_t)$ constitutes the mean of the Gaussian for both S2PG and BPTT, while $\eta_{\boldsymbol{\theta}}(s_t, z_t)$ constitutes the internal transition function for BPTT and the mean of the Gaussian distribution of the hidden state in S2PG. Figure 6 in Appendix D illustrates the policy structure. Additionally, $\Upsilon$ constitutes the covariance matrix for the Gaussian distribution of the hidden state in S2PG. Our theoretical analysis is based on prior work Papini et al. (2022); Zhao et al. (2011), which analyzes the variance of a REINFORCE-style policy gradient estimator for stateless policies and a univariate Gaussian distribution. We extend the latter to policy gradient estimators using BPTT and S2PG, and broaden the analysis to the more general multivariate Gaussian case to allow investigation on arbitrarily large action spaces. In the following, we derive upper bounds for the variance of the policy gradient for BPTT and S2PG and highlight the behavior of each estimator in different regimes. Therefore, we exploit the concept of the Frobenius norm of a matrix $\|A\|_{\mathrm{F}}$ and consider the following assumptions

**Assumption 3.1.** $r(s, a, s') \in [-R, R]$ for $R > 0$ and $\forall s_i, z_i, a_i$

**Assumption 3.2.** $\left\|\frac{\partial}{\partial \theta} f_\theta(s_i, z_i)\right\|_{\mathrm{F}} \leq F, \left\|\frac{\partial}{\partial \theta} \eta_\theta(s_i, z_i)\right\|_{\mathrm{F}} \leq H, \left\|\frac{\partial}{\partial z_i} f_\theta(s_i, z_i)\right\|_{\mathrm{F}} \leq K \quad \forall s_i, z_i$

**Assumption 3.3.** $\left\|\frac{\partial}{\partial z_i} \eta_\theta(s_i, z_i)\right\|_{\mathrm{F}} \leq Z \quad \forall s_i, z_i$

extended from Zhao et al. (2011). Then, the upper bound on the variance of BPTT is given by:

**Theorem 3.4.** *(Variance Upper Bound BPTT) Let $\nu_{\boldsymbol{\theta}}(a_t|h_t)$ be a Gaussian policy of the form $\mathcal{N}(a_t|\mu_{\boldsymbol{\theta}}^a(h_t), \Sigma)$ and let $T$ be the trajectory length. Then, under the assumptions 3.1, 3.2 and 3.3, the upper bound on the variance of the REINFORCE-style policy gradient estimate using BPTT is*

$$\mathrm{Var}\left[\nabla_\theta \hat{J}_{BPTT}(\Theta)\right] \leq \frac{R^2 \|\Sigma^{-1}\|_{\mathrm{F}} (1 - \gamma^T)^2}{N(1 - \gamma)^2} \left(TF^2 + \underbrace{2FHK\tilde{Z} + H^2 K^2 \bar{Z}}_{\Delta_{BPTT}}\right), \tag{7}$$

*where we define $\tilde{Z} = \sum_{t=0}^{T-1} \sum_{i=0}^{t-1} Z^{t-i-1}$ and $\bar{Z} = \sum_{t=0}^{T-1} \left(\sum_{i=0}^{t-1} Z^{t-i-1}\right)^2$ for brevity.*

Similarly, the upper bound on the variance of S2PG is given by:

**Theorem 3.5.** *(Variance Upper Bound S2PG) Let $\pi_{\boldsymbol{\theta}}(a_t, z_{t+1}|s_t, z_t)$ be a Gaussian policy of the form $\mathcal{N}(a_t, z_{t+1}|\mu_{\boldsymbol{\theta}}(s_t, z_t), \Sigma, \Upsilon)$ and let $T$ be the trajectory length. Then, under the assumptions 3.1 and 3.2, the upper bound on the variance of the REINFORCE-style policy gradient estimate using S2PG is*

$$\mathrm{Var}\left[\nabla_\theta \hat{J}_{S2PG}(\Theta)\right] \leq \frac{R^2 \|\Sigma^{-1}\|_{\mathrm{F}} (1 - \gamma^T)^2}{N(1 - \gamma)^2} \left(TF^2 + \underbrace{TH^2 \cdot \frac{\|\Upsilon^{-1}\|_{\mathrm{F}}}{\|\Sigma^{-1}\|_{\mathrm{F}}}}_{\Delta_{S2PG}}\right). \tag{8}$$

The proof of both Theorems can be found in Appendix A.6.1 and A.6.2. When looking at Theorem 3.4 and 3.5, it can be seen that both bounds consist of a constant factor in front and a sum over

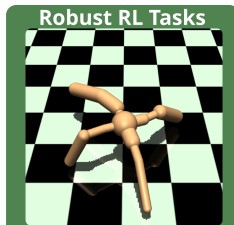
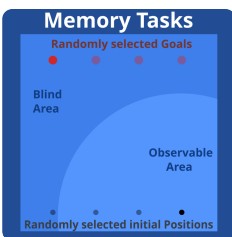
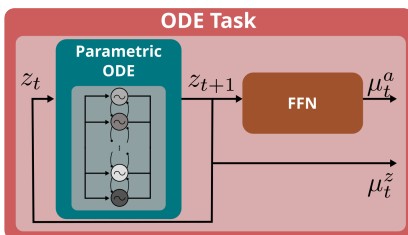
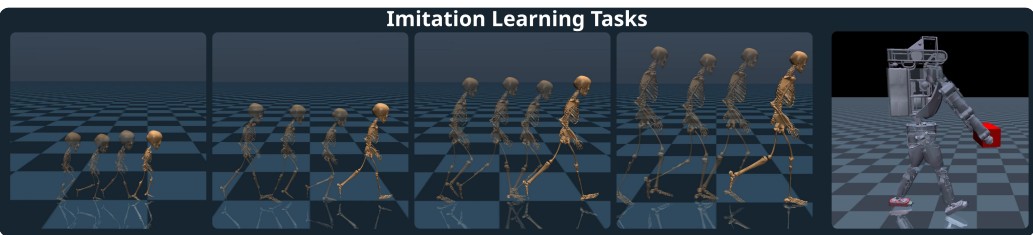

Figure 2: Overview of tasks. **Top**: From left to right, Gym locomotion tasks with randomized masses and hidden velocities, memory tasks with information hidden outside the observable area, and an ODE tasks where the policy is blind and the parameters of the ODE and a FFN are learned. **Bottom**: POMDP Humanoid walking/running task at 4.5 km/h respectively 9 km/h. The different humanoids should resemble – from left to right – an adult, a teenager ($\sim$12 years), a child ($\sim$5 years), and a toddler ($\sim$1-2 years). The target speed is scaled for the smaller humanoid according to size. The type of humanoid is hidden to the policy. A single policy is able to learn all gaits. On the right, random carry-weight task using the Atlas humanoid. The weight is randomly sampled between 0.1 kg and 10 kg at the beginning of the episode and is hidden to the policy.

$F^2$, which is a constant defining the upper bound on the gradient of $f_\theta(s_i, z_i)$. Additionally, both bounds introduce an additional term, which we consider as $\Delta$. In fact, the latter is a factor defining the additional variance added when compared to the variance of the stateless policy gradient. Hence, for $\Delta = 0$, both bounds reduce to the bounds found by prior work Papini et al. (2022); Zhao et al. (2011), when considering the univariate Gaussian case.

To understand the difference in the variance of both policy gradient estimators, we need to compare $\Delta_{BPTT}$ and $\Delta_{S2PG}$. For S2PG, the additional variance is induced by $TH^2$, which is the squared norm of the gradient of internal transition kernel $\eta_\theta(s_i, z_i)$ w.r.t. $\theta$, weighted by the ratio of the norms of the covariance matrices. In contrast, Theorem 3.4 highlights the effect of backpropagating gradients through a trajectory as $\Delta_{BPTT}$ additionally depends on the constants $\tilde{Z}$ and $\bar{Z}$. First of all, for $H \gg 1$, $K \gg 1$ and $\bar{Z} \gg 1$, we can observe that the quadratic term $H^2 K^2 \bar{Z}^2$ dominates $\Delta_{BPTT}$. Secondly, we need to distinguish two cases: as detailed in Lemma A.2 in Appendix A.6.1, for $Z \geq 1$, the term $H^2 K^2 \bar{Z}^2$ grows exponentially with the length of the trajectories $T$, indicating that exploding gradients cause exploding variance, while, for $Z < 1$, the gradients grows linearly with $T$. That is, while the variance of BPTT heavily depends on the architecture used for $\eta_\theta(s_i, z_i)$, the variance of S2PG depends much less on the architecture of the internal transition kernel allowing the use of arbitrary functions for $\eta_\theta(s_i, z_i)$ at the potential cost of higher – yet not exploding – variance. Hence, S2PG does not rely on specialized architectures that account for exploding gradients such as, GRUs or LSTMs. We note that, for S2PG, we can define a tighter bound than the one in Theorem 3.5 under the assumption of diagonal covariance matrices as shown in Lemma A.3 in Appendix A.6.2.

## 4 EXPERIMENTS

We evaluate our method across four different types of tasks. These tasks include MuJoCo Gym tasks with partial observability, both with and without a privileged critic, which we consider as Robust RL experiments, complex IL tasks with a policy under partial observability and a privileged critic, a memory task, and a task in which we train the parameters of an ODE in a policy. Figure 2 provides an overview. To conduct the evaluation, we compare recurrent versions of three popular RL algorithms—SAC, TD3, and PPO — that employ S2PG and BPTT. In Appendix C, we present the algorithm boxes for all S2PG variants. Furthermore, we compare against stateless versions of these algorithms, which use a window of the last 5 or 32 observations as the input to the policy. To show the importance of a policy state, we include basic versions of these algorithms, where only the observation is given to the policy (vanilla) and where the full state is given to the policy (oracle).

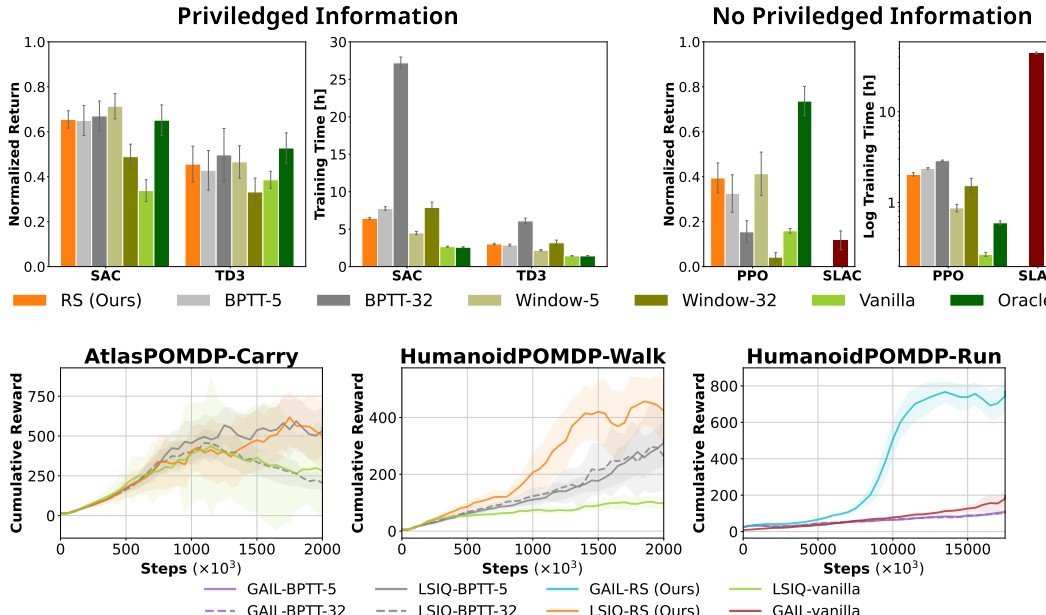

Figure 3: **Top** [Results Robust RL]: Comparison of different RL agents on POMDP tasks using our stateful gradient estimator, BPTT with a truncation length of 5 and 32, the SLAC algorithm and stateless versions of the algorithms using a window of observation with length 5 and 32. The "Oracle" approach is a vanilla version of the algorithm using full-state information. Results show the mean and 95% confidence interval of the normalized return and the training time needed for 1 million steps across all POMDP Gym tasks. **Bottom** [Results Imitation Learning]: Comparison of different versions of LS-IQ and GAIL on different imitation learning tasks under partial observability. Results show the mean and 95% confidence interval of the normalized return.

To compare with specialized algorithms for POMDPs, we compare with SLAC Lee et al. (2020a). SLAC is a model-based approach that learns complex latent variable models that are used for the critic, while the policy takes a window of observations as input. Agents that utilize our stochastic policy transition kernel are denoted by the abbreviation "RS", which stands for Recurrent Stochastic. We use Gaussian policies – for $a$ and $z'$ – in S2PG for all our experiments. The network architectures are presented in Appendix D. The initial internal state is always set to 0. When an algorithm has a replay buffer, we update its internal states when sampling transitions to resemble BPTT. For a fair comparison, all methods, except SLAC, are implemented in the same framework, MushroomRL D'Eramo et al. (2021). To properly evaluate the algorithms' training time, we allocate 4 cores on our cluster and average the results and the training time over 10 seeds for all experiments.

**Robust RL Tasks.** The first set of tasks includes the typical MuJoCo Gym locomotion tasks to test our approach in the RL setting. As done in Ni et al. (2022), we create partial observability by hiding information, specifically the velocity, from the state space. Furthermore, we randomize the mass of each link (c.f., Fig. 3). As described in Section 2.4, it is challenging to learn a stochastic policy state while using bootstrapping. Therefore, we use a privileged critic with a vanilla FFN for bootstrapping approaches. On the top-left of Figure 3, we show the cumulative returns and training times for a TD3 and a SAC-based agents. The overall results show that our approach has major computational benefits w.r.t BPTT with long histories at the price of a slight drop in asymptotic performance for SAC and TD3. We found that our approach performs better on high-dimensional observation spaces – e.g., Ant and Humanoid – and worse on low-dimensional ones. A more detailed discussion about this is given in Appendix E.4. While the window approach has computational benefits compared to BPTT, it did not perform well with increased window length. The full experimental campaign, learning curves, and more detailed task descriptions are in Appendix E.4. In the second set of tasks, labeled as 'No Privileged Information' tasks, we aim to showcase the effectiveness of our approach without using privileged information, combining the critic with Monte-Carlo rollouts. Employing the same tasks as in the privileged information setting but without mass randomization, our algorithm differs from BPTT by not utilizing a recurrent critic. The overall results, presented on the top-right of Figure 3, consistently demonstrate our method's superior performance over BPTT.

While the training time of PPO-BPTT with a truncation length of 5 is similar to our approach, notable differences emerge with longer truncation lengths. Both BPTT and the window approach exhibit worse performance with a longer history. Comparisons with specialized methods, such as SLAC Lee et al. (2020a), highlight our method's better performance and faster training. Further results, runtimes, and discussions are provided in Appendix E.3.

**Imitation Learning Tasks.** The main results of our paper are presented at the bottom of Figure 3. Here, we introduce two novel locomotion tasks. The first task is an Atlas locomotion task, where a carry-weight is randomly sampled, but the weight is hidden from the policy. The second task is a Humanoid locomotion task under dynamics randomization. The goal is to imitate a certain kinematic trajectory – either walking or running – without observing the type of the humanoid. To generate the different humanoids, we randomly sample a scaling factor. Then the links are scaled linearly, the masses are scaled cubically, the inertias are scaled quintically, and the actuator torques are scaled cubically w.r.t. the scaling factor. For both tasks, the policies only observe the positions and the velocities of the joints. Forces are not observed. As can be seen in Figure 3, our approach can be easily extended to complex IL tasks using GAIL and LS-IQ. We observe that in the IL setting our approach is able to outperform the BPTT baselines even in terms of samples. All results are given in Figure 14 and Figure 15 in Appendix E.5, where we also further discuss the results.

**Memory Task.** To show that our approach can encode longer history, we present two memory tasks. The first task is to move a point mass to a randomly sampled goal from a randomly sampled initial state. The task is shown in Figure 2. While the goal is shown to the policy at the beginning of the trajectory, it is hidden when the point mass moves away from the initial state. As can be seen in Figure 4, our approach can outperform BPTT even for longer horizons. In the second task, the positions of two doors in a maze are shown to the policy when close to the initial state and are hidden once going further away. Figure 8 provides an example and the results. As can be seen, our approach outperforms BPTT on short horizons while being slightly weaker on longer ones in the second task. We expect

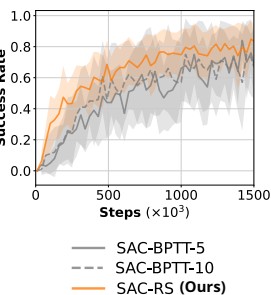

Figure 4: Point mass results.

that the reason for the drop in performance is caused by the additional variance of our method in combination with the additional absorbing states in the environment. The additional variance in our policy leads to increased exploration, which increases the chances of touching the wall and reaching an absorbing state. Nonetheless, our approach has a significantly shorter training time, analogously to the results in Figure 3. All results and a more detailed discussion are given in Appendix E.1.

**ODE Task.** Until now, all experiments were based on RNNs as stateful policies. As mentioned before, our gradient estimator could be used to train the parameters of a policy containing an arbitrary ODE. Such a policy is shown in Figure 2. This setting is particularly interesting to encode inductive biases directly into policy. To provide a proof of concept, we successfully train a completely blind policy on the Gym HalfCheetah and Ant tasks, where we used the ODE of a CPG to encode an oscillation directly into the policy; a bias that is very popular in locomotion research. The ODE is simulated using the Euler method. More results and discussions are provided in Appendix E.2.

## 5 CONCLUSIONS

This work introduced S2PG, an alternative approach for estimating the gradient of stateful policies without using BPTT by exploiting stochastic transition kernels. S2PG is easy to implement and computationally efficient. We provide a complete foundation theory of this novel estimator, allowing its implementation in state-of-the-art deep RL methods, and conduct a theoretical analysis on the variance of S2PG and BPTT. While this method still cannot replace BPTT in every setting, in the most challenging scenarios and in the IL settings, S2PG can considerably improve the performance and learning speed. Unfortunately, while S2PG can replace BPTT in the computation of the policy gradient, it is still not able to properly learn a value function and an internal policy state at the same time in the partially observable setting. While we show that our approach works well when the critic is estimated with Monte-Carlo rollouts, we limit ourselves to the setting where the critic has privileged information for bootstrapping methods. We plan to investigate solutions to this issue in the future. Finally, we will investigate how our approach scales using massively parallel environments.

ACKNOWLEDGMENTS

Calculations for this research were conducted on high-performance computers Lichtenberg and CLAIX at the NHR Centers NHR4CES at TU Darmstadt and RWTH Aachen (project numbers p0020307 & p0021606). This work was supported by the German Science Foundation (DFG) under grant number SE1042/41-1. Research presented in this paper has been partially supported by the German Federal Ministry of Education and Research (BMBF) within the subproject "Modeling and exploration of the operational area, design of the AI assistance as well as legal aspects of the use of technology" of the collaborative KIARA project (grant no. 13N16274).

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

## A PROOFS OF THEOREMS

Within this section, we refer to the following regularity conditions on the MDP taken from form Silver et al. (2014):

**Regularity Conditions A.1.** — $\iota(s,z)$, $P(s'|s,a)$, $\nabla_a P(s'|s,a)$, $\pi_{\boldsymbol{\theta}}(a, z'|s, z)$, $\nabla_{\boldsymbol{\theta}} \pi_{\theta}(a, z'|s, z)$, $\mu_{\boldsymbol{\theta}}(s, z)$, $\nabla_{\boldsymbol{\theta}} \mu_{\boldsymbol{\theta}}(s, z)$, $r(s, a)$, $\nabla_a r(s, a)$, $\iota(s)$ are continuous in all parameters and variables $s, a, z, \boldsymbol{\theta}$.

**Regularity Conditions A.2.** — There exists a $b$ and $L$ such that $\sup_s \iota(s) < b$, $\sup_{s',a,s} p(s'|s,a) < b$, $\sup_{a,s} r(s,a) < b$, $\sup_{s',a,s} \|\nabla_a p(s'|s,a)\| < L$, and $\sup_{a,s} \|\nabla_a r(s,a)\| < L$.

### A.1 STATEFUL POLICY GRADIENT WITH THE SCORE FUNCTION

*Proof.* Let

$$J(\bar{\tau}) = \sum_{t=0}^{T-1} \gamma^t r(s_t, a_t),$$

be the discounted return of a trajectory $\bar{\tau} = \langle s_0, z_0, a_1, \ldots s_{T-1}, z_{T-1}, a_{T-1}, s_T, z_T \rangle$. Differently from the standard policy gradient, we define the trajectory as a path of both the environment and the policy state. The probability of a given trajectory can be written as

$$p(\bar{\tau}|\boldsymbol{\theta}) = \iota(s_0, z_0) \prod_{t=0}^{T-1} P(s_{t+1}|s_t, a_t) \pi_{\boldsymbol{\theta}}(a_t, z_{t+1}|s_t, z_t).$$

The continuity of $\iota(s_0, z_0)$, $P(s_{t+1}|s_t, a_t)$, $\pi_{\boldsymbol{\theta}}(a_t, z_{t+1}|s_t, z_t)$, $\nabla_{\boldsymbol{\theta}} \pi_{\boldsymbol{\theta}}(a_t, z_{t+1}|s_t, z_t)$ implies that $p(\bar{\tau}|\boldsymbol{\theta})$ and $\nabla_{\boldsymbol{\theta}} p(\bar{\tau}|\boldsymbol{\theta})$ are continue as well. Hence, we can apply the Leibniz rule to exchange the order of integration and derivative, allowing us to follow the same steps as the classical policy gradient

$$\nabla_{\boldsymbol{\theta}} \mathcal{J}(\pi_{\boldsymbol{\theta}}) = \nabla_{\boldsymbol{\theta}} \int p(\bar{\tau}|\boldsymbol{\theta}) J(\bar{\tau}) d\bar{\tau} = \int \nabla_{\boldsymbol{\theta}} p(\bar{\tau}|\boldsymbol{\theta}) J(\bar{\tau}) d\bar{\tau}.$$

Applying the likelihood ratio trick, we obtain

$$\nabla_{\boldsymbol{\theta}} \mathcal{J}(\pi_{\boldsymbol{\theta}}) = \int p(\bar{\tau}|\boldsymbol{\theta}) \nabla_{\boldsymbol{\theta}} \log p(\bar{\tau}|\boldsymbol{\theta}) J(\bar{\tau}) d\bar{\tau} = \mathbb{E}_{\bar{\tau}} \left[ \nabla_{\boldsymbol{\theta}} \log p(\bar{\tau}|\boldsymbol{\theta}) J(\bar{\tau}) \right]. \tag{9}$$

To obtain the REINFORCE-style estimator, we note that

$$\log p(\bar{\tau}|\boldsymbol{\theta}) = \log \iota(s_0, z_0) + \sum_{t=0}^{T-1} \log P(s_{t+1}|s_t, a_t) + \sum_{t=0}^{T-1} \log \pi_{\boldsymbol{\theta}}(a_t, z_{t+1}|s_t, z_t).$$

The gradient $\nabla_{\boldsymbol{\theta}} p(\bar{\tau}|\boldsymbol{\theta})$ only depends on the policy and not the initial state distribution or the state transitions. Therefore, we can write the stateful policy gradient as

$$\nabla_{\boldsymbol{\theta}} \mathcal{J}(\pi_{\boldsymbol{\theta}}) = \mathbb{E}_{\tau} \left[ \sum_{t=0}^{T-1} \nabla_{\boldsymbol{\theta}} \log \pi_{\boldsymbol{\theta}}(a_t, z_{t+1}|s_t, z_t) J(\tau) \right], \tag{10}$$

concluding the proof. $\qquad\qquad\square$

### A.2 STATEFUL POLICY GRADIENT IN PARTIALLY OBSERVABLE ENVIRONMENTS

*Proof.* The derivation follows exactly the one for the fully observable scenario, with the difference that we need to deal with the observations instead of states. The probability of observing a trajectory $\bar{\tau}$ in a POMDP is

$$p(\bar{\tau}|\boldsymbol{\theta}) = \iota(s_0, z_0) O(o_0|s_0) \prod_{t=0}^{T-1} O(o_{t+1}|s_{t+1}) P(s_{t+1}|s_t, a_t) \pi_{\boldsymbol{\theta}}(a_t, z_{t+1}|o_t, z_t).$$

Computing the term $\nabla_{\boldsymbol{\theta}} \log p(\bar{\tau}|\boldsymbol{\theta})$ and replacing it into equation 9 we obtain

$$\nabla_{\boldsymbol{\theta}} \mathcal{J}(\pi_{\boldsymbol{\theta}}) = \mathbb{E}_{\bar{\tau}} \left[ \sum_{t=0}^{T-1} \nabla_{\boldsymbol{\theta}} \log \pi_{\boldsymbol{\theta}}(a_t, z_{t+1}|o_t, z_t) J(\bar{\tau}) \right],$$

using the same simplification as done in equation 10, concluding the proof. $\qquad\square$

### A.3 STATEFUL POLICY GRADIENT THEOREM

*Proof.* We can write the objective function as follows

$$\mathcal{J}_{\boldsymbol{\theta}} = \mathop{\mathbb{E}}_{(s_0, z_0) \sim \iota} [V_{\pi_{\boldsymbol{\theta}}}(s_0, z_0)] = \mathop{\mathbb{E}}_{\substack{(s_0, z_0) \sim \iota, \\ (a_0, z_1) \sim \pi_{\boldsymbol{\theta}}}} [Q^{\pi_{\boldsymbol{\theta}}}(s_0, z_0, a_0, z_1)].$$

We can now compute the gradient as

$$\begin{aligned}
\nabla_{\boldsymbol{\theta}} \mathcal{J}_{\boldsymbol{\theta}} &= \nabla_{\boldsymbol{\theta}} \mathop{\mathbb{E}}_{\substack{(s_0, z_0) \sim \iota, \\ (a_0, z_1) \sim \pi_{\boldsymbol{\theta}}}} [Q^{\pi_{\boldsymbol{\theta}}}(s_0, z_0, a_0, z_1)] \\
&= \nabla_{\boldsymbol{\theta}} \int_{\mathcal{S}} \int_{\mathcal{Z}} \int_{\mathcal{A}} \int_{\mathcal{Z}} \iota(s_0, z_0) \pi_{\boldsymbol{\theta}}(a_0, z_1 | s_0, z_0) Q^{\pi_{\boldsymbol{\theta}}}(s_0, z_0, a_0, z_1) da_0 dz_1 ds_0 dz_0.
\end{aligned}$$

The regularity conditions A.1 imply that $Q^{\pi_{\boldsymbol{\theta}}}(s, z', a, z)$ and $\nabla_{\boldsymbol{\theta}} Q^{\pi_{\boldsymbol{\theta}}}(s, z', a, z)$ are continuous functions of $\boldsymbol{\theta}$, $s$, $a$, and $z$. Further, the compactness of $\mathcal{S}$, $\mathcal{A}$ and $\mathcal{Z}$ implies that for any $\boldsymbol{\theta}$, $\|\nabla_{\boldsymbol{\theta}} Q^{\pi_{\boldsymbol{\theta}}}(s, z', a, z)\|$ and $\|\nabla_{\boldsymbol{\theta}} \pi_{\boldsymbol{\theta}}(a, z' | s, z)\|$ are bounded functions of $s$, $a$ and $z$. These conditions are required to exchange integration and derivatives, as well as the order of integration throughout the this proof. Hence, the gradient can be written as

$$\begin{aligned}
\nabla_{\boldsymbol{\theta}} \mathcal{J}_{\boldsymbol{\theta}} &= \int_{\mathcal{S}} \int_{\mathcal{Z}} \int_{\mathcal{A}} \int_{\mathcal{Z}} \nabla_{\boldsymbol{\theta}} \left[ \iota(s_0, z_0) \pi_{\boldsymbol{\theta}}(a_0, z_1 | s_0, z_0) Q^{\pi_{\boldsymbol{\theta}}}(s_0, z_0, a_0, a_1) \right] da_0 dz_1 ds_0 dz_0 \\
&= \int_{\mathcal{S}} \int_{\mathcal{Z}} \int_{\mathcal{A}} \int_{\mathcal{Z}} \iota(s_0, z_0) \nabla_{\boldsymbol{\theta}} \pi_{\boldsymbol{\theta}}(a_0, z_1 | s_0, z_0) Q^{\pi_{\boldsymbol{\theta}}}(s_0, z_0, a_0, z_1) \\
&\qquad + \iota(s_0, z_0) \pi_{\boldsymbol{\theta}}(a_0, z_1 | s_0, z_0) \nabla_{\boldsymbol{\theta}} Q^{\pi_{\boldsymbol{\theta}}}(s_0, z_0, a_0, z_1) da_0 dz_1 ds_0 dz_0.
\end{aligned}$$

We now focus on the gradient of the Q-function w.r.t. the policy parameters. Using the following relationship

$$Q^{\pi_{\boldsymbol{\theta}}}(s_t, z_t, a_t, z_{t+1}) = r(s_t, a_t) + \gamma \int_{\mathcal{S}} P(s_{t+1} | s_t, a_t) V^{\pi_{\boldsymbol{\theta}}}(s_{t+1}, z_{t+1}) ds_{t+1}.$$

We can write

$$\begin{aligned}
\nabla_{\boldsymbol{\theta}} Q^{\pi_{\boldsymbol{\theta}}}(s_t, z_t, a_t, z_{t+1}) &= \nabla_{\boldsymbol{\theta}} \left[ r(s_t, a_t) + \gamma \int_{\mathcal{S}} P(s_{t+1} | s_t, a_t) V^{\pi_{\boldsymbol{\theta}}}(s_{t+1}, z_{t+1}) ds_{t+1} \right] \\
&= \gamma \int_{\mathcal{S}} P(s_{t+1} | s_t, a_t) \nabla_{\boldsymbol{\theta}} V^{\pi_{\boldsymbol{\theta}}}(s_{t+1}, z_{t+1}) ds_{t+1}.
\end{aligned}$$

Similarly, we can compute the gradient of the value function as

$$\begin{aligned}
\nabla_{\boldsymbol{\theta}} V^{\pi_{\boldsymbol{\theta}}}(s_t, z_t) &= \int_{\mathcal{A}} \int_{\mathcal{Z}} \Big( \nabla_{\boldsymbol{\theta}} \pi_{\boldsymbol{\theta}}(a_t, z_{t+1} | s_t, z_t) Q^{\pi_{\boldsymbol{\theta}}}(s_t, z_t, a_t, z_{t+1}) \\
&\qquad + \gamma \int_{\mathcal{S}} \pi_{\boldsymbol{\theta}}(a_t, z_{t+1} | s_t, z_t) P(s_{t+1} | s_t, a_t) \nabla_{\boldsymbol{\theta}} V^{\pi_{\boldsymbol{\theta}}}(s_{t+1}, z_{t+1}) \Big) ds_{t+1} da_t dz_{t+1}.
\end{aligned}$$

We now focus again on $\nabla_{\boldsymbol{\theta}} \mathcal{J}_{\boldsymbol{\theta}}$. Using the linearity of integrals, we write

$$\begin{aligned}
\nabla_{\boldsymbol{\theta}} \mathcal{J}_{\boldsymbol{\theta}} &= \int_{\mathcal{S}} \int_{\mathcal{Z}} \int_{\mathcal{A}} \int_{\mathcal{Z}} \iota(s_0, z_0) \nabla_{\boldsymbol{\theta}} \pi_{\boldsymbol{\theta}}(a_0, z_1 | s_0, z_0) Q^{\pi_{\boldsymbol{\theta}}}(s_0, z_0, a_0, z_1) da_0 dz_1 ds_0 dz_0 \\
&\quad + \int_{\mathcal{S}} \int_{\mathcal{Z}} \int_{\mathcal{A}} \int_{\mathcal{Z}} \iota(s_0, z_0) \pi_{\boldsymbol{\theta}}(a_0, z_1 | s_0, z_0) \nabla_{\boldsymbol{\theta}} Q^{\pi_{\boldsymbol{\theta}}}(s_0, z_0, a_0, z_1) da_0 dz_1 ds_0 dz_0.
\end{aligned}$$

Now, using the expressions for $\nabla_{\boldsymbol{\theta}} V^{\pi_{\boldsymbol{\theta}}}$ and $\nabla_{\boldsymbol{\theta}} Q^{\pi_{\boldsymbol{\theta}}}$, we can expand the second term of the previous sum for one step (limited to one step to highlight the structure)

$$
\int_{\mathcal{S}} \int_{\mathcal{Z}} \int_{\mathcal{A}} \int_{\mathcal{Z}} \iota(s_0, z_0) \pi_{\boldsymbol{\theta}}(a_0, z_0 | s_0, z_0) \nabla_{\boldsymbol{\theta}} Q^{\pi_{\boldsymbol{\theta}}}(s_0, z_0, a_0, z_1) da_0 dz_1 ds_0 dz_0
$$

$$
= \gamma \int_{\mathcal{S}} \int_{\mathcal{Z}} \int_{\mathcal{A}} \int_{\mathcal{Z}} \int_{\mathcal{S}} \iota(s_0, z_0) \pi_{\boldsymbol{\theta}}(a_0, z_0 | s_0) P(s_1 | s_0, a_0) \nabla_{\boldsymbol{\theta}} V^{\pi_{\boldsymbol{\theta}}}(s_1, z_1) ds_1 da_0 dz_1 ds_0 dz_0
$$

$$
= \gamma \int_{\mathcal{S}} \int_{\mathcal{Z}} \int_{\mathcal{A}} \int_{\mathcal{Z}} \int_{\mathcal{S}} \iota(s_0, z_0) \pi_{\boldsymbol{\theta}}(a_0, z_1 | s_0, z_0) P(s_1 | s_0, a_0) \cdot
$$

$$
\cdot \left[ \int_{\mathcal{A}} \int_{\mathcal{Z}} \nabla_{\boldsymbol{\theta}} \pi_{\boldsymbol{\theta}}(a_1, z_2 | s_1, z_1) Q^{\pi_{\boldsymbol{\theta}}}(s_1, z_1, a_1, z_2) \right.
$$

$$
\left. + \gamma \int_{\mathcal{S}} \pi_{\boldsymbol{\theta}}(a_1, z_2 | s_1, z_1) P(s_2 | s_1, a_1) \nabla_{\boldsymbol{\theta}} V^{\pi_{\boldsymbol{\theta}}}(s_2, z_2) ds_2 da_1 dz_2 \right] ds_1 da_0 dz_1 ds_0 dz_0.
$$

By splitting the integrals again and rearranging the terms, we obtain

$$
\int_{\mathcal{S}} \int_{\mathcal{Z}} \int_{\mathcal{A}} \int_{\mathcal{Z}} \iota(s_0, z_0) \pi_{\boldsymbol{\theta}}(a_0, z_1 | s_0, z_0) \nabla_{\boldsymbol{\theta}} Q^{\pi_{\boldsymbol{\theta}}}(s_0, z_0, a_0, z_1) da_0 dz_1 ds_0 dz_0 =
$$

$$
\gamma \int_{\mathcal{S}} \int_{\mathcal{Z}} \int_{\mathcal{A}} \int_{\mathcal{Z}} \int_{\mathcal{S}} \iota(s_0, z_0) \pi_{\boldsymbol{\theta}}(a_0, z_1 | s_0, z_0) P(s_1 | s_0, a_0) da_0 dz_1 ds_0 dz_0 \cdot
$$

$$
\cdot \left[ \int_{\mathcal{A}} \int_{\mathcal{Z}} \nabla_{\boldsymbol{\theta}} \pi_{\boldsymbol{\theta}}(a_1, z_2 | s_1, z_1) Q^{\pi_{\boldsymbol{\theta}}}(s_1, z_1, a_1, z_2) da_1 dz_2 \right] ds_1
$$

$$
+ \gamma^2 \int_{\mathcal{S}} \int_{\mathcal{Z}} \int_{\mathcal{A}} \int_{\mathcal{Z}} \int_{\mathcal{S}} \int_{\mathcal{A}} \int_{\mathcal{Z}} \int_{\mathcal{S}} \iota(s_0, z_0) \pi_{\boldsymbol{\theta}}(a_0, z_1 | s_0, z_0) P(s_1 | s_0, a_0) \pi_{\boldsymbol{\theta}}(a_1, z_2 | s_1, z_1) P(s_2 | s_1, a_1) \cdot
$$

$$
\cdot \nabla_{\boldsymbol{\theta}} V^{\pi_{\boldsymbol{\theta}}}(s_2, z_2) ds_2 da_1 dz_2 ds_1 da_0 dz_1 ds_0 dz_0.
$$

To proceed and simplify the notation, we introduce the notion of t-step transition density $\mathrm{Tr}_t^{\pi_{\boldsymbol{\theta}}}$, i.e. the density function of transitioning to a given state and policy state in $t$ steps under the stochastic policy $\pi_{\boldsymbol{\theta}}$. In particular, we can write

$$
\mathrm{Tr}_0^{\pi_{\boldsymbol{\theta}}}(s_0, z_0) = \iota(s_0, z_0),
$$

$$
\mathrm{Tr}_1^{\pi_{\boldsymbol{\theta}}}(s_1, z_1) = \int_{\mathcal{A}} \int_{\mathcal{Z}} \int_{\mathcal{S}} \iota(s_0, z_0) \pi_{\boldsymbol{\theta}}(a_0, z_1 | s_0, z_0) P(s_1 | s_0, a_0) ds_0 dz_0 da_0,
$$

$$
\mathrm{Tr}_t^{\pi_{\boldsymbol{\theta}}}(s_t, z_t) = \int_{\mathcal{S}^t \times \mathcal{Z}^t \times \mathcal{A}^t} \iota(s_0, z_0) \prod_{k=0}^{t-1} \pi_{\boldsymbol{\theta}}(a_k, z_{k+1} | s_k, z_k) P(s_{k+1} | s_k, a_k) ds_k dz_k da_k.
$$

Now we can put everything together, expanding the term containing $\nabla_{\boldsymbol{\theta}} V^{\pi_{\boldsymbol{\theta}}}$ an infinite amount of times

$$
\nabla_{\boldsymbol{\theta}} \mathcal{J}_{\boldsymbol{\theta}} = \int_{\mathcal{S}} \int_{\mathcal{Z}} \mathrm{Tr}_0^{\pi_{\boldsymbol{\theta}}}(s_0, z_0) \int_{\mathcal{A}} \int_{\mathcal{Z}} \nabla_{\boldsymbol{\theta}} \pi_{\boldsymbol{\theta}}(a_0, z_1 | s_0, z_0) Q^{\pi_{\boldsymbol{\theta}}}(s_0, z_0, a_0, z_1) da_0 dz_1 ds_0 dz_0
$$

$$
+ \gamma \int_{\mathcal{S}} \int_{\mathcal{Z}} \mathrm{Tr}_1^{\pi_{\boldsymbol{\theta}}}(s_1, z_1) \int_{\mathcal{A}} \int_{\mathcal{Z}} \nabla_{\boldsymbol{\theta}} \pi_{\boldsymbol{\theta}}(a_1, z_2 | s_1, z_1) Q^{\pi_{\boldsymbol{\theta}}}(s_1, z_1, a_1, z_2) da_1 dz_2 ds_1 dz_1
$$

$$
+ \gamma^2 \int_{\mathcal{S}} \int_{\mathcal{Z}} \mathrm{Tr}_2^{\pi_{\boldsymbol{\theta}}}(s_2, z_2) \int_{\mathcal{A}} \int_{\mathcal{Z}} \nabla_{\boldsymbol{\theta}} \pi_{\boldsymbol{\theta}}(a_2, z_3 | s_2, z_2) Q^{\pi_{\boldsymbol{\theta}}}(s_2, z_2, a_2, z_3) da_2 dz_3 ds_2 dz_2
$$

$$
+ \gamma^3 \int_{\mathcal{S}} \int_{\mathcal{Z}} \mathrm{Tr}_3^{\pi_{\boldsymbol{\theta}}}(s_3, z_3) \int_{\mathcal{A}} \int_{\mathcal{Z}} \nabla_{\boldsymbol{\theta}} \pi_{\boldsymbol{\theta}}(a_3, z_4 | s_3, z_3) Q^{\pi_{\boldsymbol{\theta}}}(s_3, z_3, a_3, z_4) da_3 dz_4 ds_3 dz_4
$$

$$
+ \dots
$$

$$
= \sum_{t=0}^{\infty} \gamma^t \int_{\mathcal{S}} \int_{\mathcal{Z}} \mathrm{Tr}_t^{\pi_{\boldsymbol{\theta}}}(s_t, z_t) \cdot
$$

$$
\cdot \int_{\mathcal{A}} \int_{\mathcal{Z}} \nabla_{\boldsymbol{\theta}} \pi_{\boldsymbol{\theta}}(a_t, z_{t+1} | s_t, z_t) Q^{\pi_{\boldsymbol{\theta}}}(s_t, z_t, a_t, z_{t+1}) da_t dz_{t+1} ds_t dz_t.
$$

Exchanging the integral and the series we obtain

$$\nabla_{\boldsymbol{\theta}}\mathcal{J}_{\boldsymbol{\theta}} = \int_{\mathcal{S}}\int_{\mathcal{Z}}\sum_{t=0}^{\infty}\gamma^t \mathrm{Tr}_t^{\pi_{\boldsymbol{\theta}}}(s_t, z_t) \cdot$$
$$\int_{\mathcal{A}}\int_{\mathcal{Z}}\nabla_{\boldsymbol{\theta}}\pi_{\boldsymbol{\theta}}(a_t, z_{t+1}|s_t, z_t)Q^{\pi_{\boldsymbol{\theta}}}(s_t, z_t, a_t, z_{t+1})da_t dz_{t+1}ds_t dz_t.$$

Now, with the definition of the occupancy metric

$$\rho^{\pi_{\boldsymbol{\theta}}}(s, z) = \sum_{t=0}^{\infty}\gamma^t \mathrm{Tr}_t^{\pi_{\boldsymbol{\theta}}}(s, z),$$

and by relabelling $s_t$, $z_t$, $a_t$, $z_{t+1}$ into $s$, $z$, $a$ and $z'$ we obtain

$$\nabla_{\boldsymbol{\theta}}\mathcal{J}_{\boldsymbol{\theta}} = \int_{\mathcal{S}}\int_{\mathcal{Z}}\rho^{\pi_{\boldsymbol{\theta}}}(s, z)\int_{\mathcal{A}}\int_{\mathcal{Z}}\nabla_{\boldsymbol{\theta}}\pi_{\boldsymbol{\theta}}(a, z'|s, z)\,Q^{\pi_{\boldsymbol{\theta}}}(s, z, a, z')da\,dz'ds\,dz.$$

$\square$

## A.4 Stateful Performance Difference Lemma

This lemma is particularly useful because it allows using the advantage function $A^q$ computed for an arbitrary policy $q$ to evaluate the performance of our parametric policy $\pi_{\boldsymbol{\theta}}$. As $q$ is an arbitrary policy not depending on $\boldsymbol{\theta}$, the left-hand side of the Performance difference lemma can be used as a surrogate loss. This lemma allows to easily derive S2PG version of PPO and TRPO.

**Lemma A.1.** *(**Performance Difference Lemma for Stateful Policies**) For any stateful policy $\pi$ and $q$ in an arbitrary MDP, the difference of performance of the two policies in terms of expected discounted return can be computed as*

$$\mathcal{J}(\pi) - \mathcal{J}(q) = \mathop{\mathbb{E}}_{\bar{\tau}\sim\pi,P}\left[\sum_{t=0}^{\infty}\gamma^t A^q(s_t, z_t, a_t, z_{t+1})\right]$$

*Proof.* Starting from the lemma

$$\mathcal{J}(\pi) - \mathcal{J}(q) = \mathop{\mathbb{E}}_{\bar{\tau}\sim\pi,P}\left[\sum_{t=0}^{\infty}\gamma^t A^q(s_t, z_t, a_t, z_{t+1})\right],$$

we can rewrite the advantage as

$$A^q(s_t, z_t, a_t, z_{t+1}) = \mathop{\mathbb{E}}_{s_{t+1}\sim P}\left[r(s_t, a_t) + \gamma V^q(s_{t+1}, z_{t+1}) - V^q(s_t, z_t)\right].$$

We can then manipulate the right-hand side as follows

$$\mathop{\mathbb{E}}_{\bar{\tau}\sim\pi,P}\left[\sum_{t=0}^{\infty}\gamma^t A^q(s_t, z_t, a_t, z_{t+1})\right] = \mathop{\mathbb{E}}_{\bar{\tau}\sim\pi,P}\left[\sum_{t=0}^{\infty}\gamma^t\left(r(s_t, a_t) + \gamma V^q(s_{t+1}, z_{t+1}) - V^q(s_t, z_t)\right)\right]$$
$$= \mathop{\mathbb{E}}_{\bar{\tau}\sim\pi,P}\left[\sum_{t=0}^{\infty}\gamma^t r(s_t, a_t)\right]$$
$$+ \mathop{\mathbb{E}}_{\bar{\tau}\sim\pi,P}\left[\sum_{t=0}^{\infty}\gamma^{t+1}V^q(s_{t+1}, z_{t+1}) - \sum_{t=0}^{\infty}\gamma^t V^q(s_t, z_t)\right]$$
$$= \mathcal{J}(\pi) + \mathop{\mathbb{E}}_{\bar{\tau}\sim\pi,P}\left[\sum_{t=1}^{\infty}\gamma^t V^q(s_t, z_t) - \sum_{t=0}^{\infty}\gamma^t V^q(s_t, z_t)\right]$$
$$= \mathcal{J}(\pi) + \mathop{\mathbb{E}}_{(s,z)\sim\iota}\left[V^q(s)\right]$$
$$= \mathcal{J}(\pi) - \mathcal{J}(q),$$

concluding the proof. $\square$

A.5  STATEFUL DETERMINISTIC POLICY GRADIENT THEOREM

*Proof.* The proof proceeds in a very similar fashion as the one for the standard DPGT. We start by computing the gradient of a value function at an arbitrary state $s$ and an arbitrary policy state $z$

$$
\begin{aligned}
\nabla_{\boldsymbol{\theta}} V^{\mu_{\boldsymbol{\theta}}}(s,z) =& \nabla_{\boldsymbol{\theta}} Q^{\mu_{\boldsymbol{\theta}}}(s, z, \mu_{\boldsymbol{\theta}}^a(s,z), \mu_{\boldsymbol{\theta}}^z(s,z)) \\
=& \nabla_{\boldsymbol{\theta}} \left( r(s, \mu_{\boldsymbol{\theta}}^a(s,z)) + \int_{\mathcal{S}} \gamma P(s'|s, \mu_{\boldsymbol{\theta}}^a(s,z)) V^{\mu_{\boldsymbol{\theta}}}(s', \mu_{\boldsymbol{\theta}}^z(s,z)) ds' \right) \\
=& \nabla_{\boldsymbol{\theta}} \mu_{\boldsymbol{\theta}}^a(s,z) \nabla_a r(s,a)|_{a=\mu_{\boldsymbol{\theta}}^a(s,z)} + \nabla_{\boldsymbol{\theta}} \int_{\mathcal{S}} \gamma P(s'|s, \mu_{\boldsymbol{\theta}}^a(s,z)) V^{\mu_{\boldsymbol{\theta}}}(s', \mu_{\boldsymbol{\theta}}^z(s,z)) ds' \\
=& \nabla_{\boldsymbol{\theta}} \mu_{\boldsymbol{\theta}}^a(s,z) \nabla_a r(s,a)|_{a=\mu_{\boldsymbol{\theta}}^a(s,z)} + \int_{\mathcal{S}} \gamma \Big( P(s'|s, \mu_{\boldsymbol{\theta}}^a(s,z)) \nabla_{\boldsymbol{\theta}} V^{\mu_{\boldsymbol{\theta}}}(s', \mu_{\boldsymbol{\theta}}^z(s,z)) \\
& + \nabla_{\boldsymbol{\theta}} \mu_{\boldsymbol{\theta}}^a(s,z) \nabla_a P(s'|s,a)|_{a=\mu_{\boldsymbol{\theta}}^a(s,z)} V(s, \mu_{\boldsymbol{\theta}}^z(s,z)) \Big) ds' \\
=& \nabla_{\boldsymbol{\theta}} \mu_{\boldsymbol{\theta}}^a(s,z) \nabla_a \left( r(s,a) + \int_{\mathcal{S}} \gamma P(s'|s,a) V^{\mu_{\boldsymbol{\theta}}}(s', \mu_{\boldsymbol{\theta}}^z(s,z)) ds' \right) \Big|_{a=\mu_{\boldsymbol{\theta}}^a(s,z)} \\
& + \nabla_{\boldsymbol{\theta}} \mu_{\boldsymbol{\theta}}^z(s,z) \left( \int_{\mathcal{S}} \gamma P(s'|s,a)|_{a=\mu_{\boldsymbol{\theta}}^a(s,z)} \nabla_{z'} V^{\mu_{\boldsymbol{\theta}}}(s', z') ds' \right) \Big|_{z'=\mu_{\boldsymbol{\theta}}^z(s,z)} \\
& + \int_{\mathcal{S}} \gamma P(s'|s, \mu_{\boldsymbol{\theta}}^a(s,z)) \left. \nabla_{\boldsymbol{\theta}} V^{\mu_{\boldsymbol{\theta}}}(s', z') \right|_{z'=\mu_{\boldsymbol{\theta}}^z(s,z)} ds' \,.
\end{aligned}
\tag{11}
$$

Now, we can observe that

$$
\begin{aligned}
\nabla_{z'} Q^{\mu_{\boldsymbol{\theta}}}(s, z, a, z') &= \nabla_{z'} \left( r(s,a) + \int_{\mathcal{S}} \gamma P(s'|s,a) V^{\mu_{\boldsymbol{\theta}}}(s', z') ds' \right) \\
&= \int_{\mathcal{S}} \gamma P(s'|s,a) \nabla_{z'} V^{\mu_{\boldsymbol{\theta}}}(s', z') ds' \,.
\end{aligned}
\tag{12}
$$

Substituting equation 12 into equation 11 yields

$$
\begin{aligned}
\nabla_{\boldsymbol{\theta}} V^{\mu_{\boldsymbol{\theta}}}(s,z) =& \nabla_{\boldsymbol{\theta}} \mu_{\boldsymbol{\theta}}^a(s,z) \nabla_a \left. Q^{\mu_{\boldsymbol{\theta}}}(s, z, a, \mu_{\boldsymbol{\theta}}^z(s,z)) \right|_{a=\mu_{\boldsymbol{\theta}}^a(s,z)} \\
& + \nabla_{\boldsymbol{\theta}} \mu_{\boldsymbol{\theta}}^z(s,z) \left. \nabla_{z'} Q^{\mu_{\boldsymbol{\theta}}}(s, z, \mu_{\boldsymbol{\theta}}^a(s,z), z') \right|_{z'=\mu_{\boldsymbol{\theta}}^z(s,z)} \\
& + \int_{\mathcal{S}} \gamma P(s'|s, \mu_{\boldsymbol{\theta}}^a(s,z)) \left. \nabla_{\boldsymbol{\theta}} V^{\mu_{\boldsymbol{\theta}}}(s', z') \right|_{z'=\mu_{\boldsymbol{\theta}}^z(s,z)} ds' \,.
\end{aligned}
\tag{13}
$$

To simplify the notation we write

$$
\begin{aligned}
G^{\mu_{\boldsymbol{\theta}}}(s,z) =& \nabla_{\boldsymbol{\theta}} \mu_{\boldsymbol{\theta}}^a(s,z) \nabla_a \left. Q^{\mu_{\boldsymbol{\theta}}}(s, z, a, \mu_{\boldsymbol{\theta}}^z(s,z)) \right|_{a=\mu_{\boldsymbol{\theta}}^a(s,z)} \\
& + \nabla_{\boldsymbol{\theta}} \mu_{\boldsymbol{\theta}}^z(s,z) \left. \nabla_{z'} Q^{\mu_{\boldsymbol{\theta}}}(s, z, \mu_{\boldsymbol{\theta}}^a(s,z), z') \right|_{z'=\mu_{\boldsymbol{\theta}}^z(s,z)} \,.
\end{aligned}
$$

Therefore, equation 13 can be written compactly as

$$
\nabla_{\boldsymbol{\theta}} V^{\mu_{\boldsymbol{\theta}}}(s,z) = G^{\mu_{\boldsymbol{\theta}}}(s,z) + \int_{\mathcal{S}} \gamma P(s'|s, \mu_{\boldsymbol{\theta}}^a(s,z)) \left. \nabla_{\boldsymbol{\theta}} V^{\mu_{\boldsymbol{\theta}}}(s', z') \right|_{z'=\mu_{\boldsymbol{\theta}}^z(s,z)} ds' \,.
\tag{14}
$$

Using equation 14, we can write the gradient of the objective function as

$$
\begin{aligned}
\nabla_{\boldsymbol{\theta}} \mathcal{J}(\mu_{\boldsymbol{\theta}}) =& \nabla_{\boldsymbol{\theta}} \int_{\mathcal{S}} \int_{\mathcal{Z}} \iota(s_0, z_0) V^{\mu_{\boldsymbol{\theta}}}(s_0, z_0) ds_0 dz_0 \\
=& \int_{\mathcal{S}} \int_{\mathcal{Z}} \iota(s_0, z_0) G^{\mu_{\boldsymbol{\theta}}}(s_0, z_0) ds_0 dz_0 \\
& + \int_{\mathcal{S}} \int_{\mathcal{Z}} \iota(s_0, z_0) \int_{\mathcal{S}} \gamma P(s_1|s_0, \mu_{\boldsymbol{\theta}}^a(s_0, z_0)) \left. \nabla_{\boldsymbol{\theta}} V^{\mu_{\boldsymbol{\theta}}}(s_1, z_1) \right|_{z_1=\mu_{\boldsymbol{\theta}}^z(s_0, z_0)} ds_1 ds_0 dz_0 \,.
\end{aligned}
\tag{15}
$$

Before continuing the proof, we need to notice that the variables $s'$ and $z'$ are conditionally independent if the previous states $s$ and $z$ are given. Furthermore, it is important to notice that, even if the

policy state $z$ is a deterministic function of the previous state and previous policy state, the variable $z$ is still a random variable if we consider its value after more than two steps of the environment, as its value depends on a set of random variables, namely the states encountered during the path. Using these two assumptions, we extract the joint occupancy measure of $s$ and $z$. Unfortunately, to proceed with the proof we need to abuse the notation (as commonly done in engineering) and use the Dirac's delta distribution $\delta(x)$.

Due to conditional independence, we can write the joint transition probability under the deterministic policy $\mu_{\boldsymbol{\theta}}$ as

$$P^{\mu_{\boldsymbol{\theta}}}(s', z'|s, z) = P^{\mu_{\boldsymbol{\theta}}}(s'|s, z)\delta^{\mu_{\boldsymbol{\theta}}}(z'|s, z) = P(s'|s, \mu_{\boldsymbol{\theta}}^a(s, z))\delta(z' - \mu_{\boldsymbol{\theta}}^z(s, z)),$$

with $P^{\mu_{\boldsymbol{\theta}}}(s'|s, z) = P(s'|s, \mu_{\boldsymbol{\theta}}^a(s, z))$ and $\delta^{\mu_{\boldsymbol{\theta}}}(z'|s, z) = \delta(z' - \mu_{\boldsymbol{\theta}}^z(s, z))$.

Now we can write

$$P^{\mu_{\boldsymbol{\theta}}}(s'|s, z) = \int_{\mathcal{Z}} P^{\mu_{\boldsymbol{\theta}}}(s', z'|s, z)dz' = \int_{\mathcal{Z}} P^{\mu_{\boldsymbol{\theta}}}(s'|s, z)\delta^{\mu_{\boldsymbol{\theta}}}(z'|s, z)dz'.$$

We introduce, as done for the PGT, the t-steps transition density $\mathrm{Tr}_t^{\mu_{\boldsymbol{\theta}}}$ under the deterministic policy $\mu_{\boldsymbol{\theta}}$

$$\mathrm{Tr}_0^{\mu_{\boldsymbol{\theta}}}(s_0, z_0) = \iota(s_0, z_0),$$

$$\mathrm{Tr}_1^{\mu_{\boldsymbol{\theta}}}(s_1, z_1) = \int_{\mathcal{S}}\int_{\mathcal{Z}} \iota(s_0, z_0)P^{\mu_{\boldsymbol{\theta}}}(s_1|s_0, z_0)\delta^{\mu_{\boldsymbol{\theta}}}(z_1|s_0, z_0)ds_0 dz_0,$$

$$\mathrm{Tr}_t^{\mu_{\boldsymbol{\theta}}}(s_t, z_t) = \int_{\mathcal{S}^t \times \mathcal{Z}^t} \iota(s_0, z_0)\prod_{k=0}^{t-1} P^{\mu_{\boldsymbol{\theta}}}(s_{k+1}|s_k, z_k)\delta^{\mu_{\boldsymbol{\theta}}}(z_{k+1}|s_k, z_k)ds_k dz_k.$$

Using the previously introduced notation, we can expand equation 15, by recursion, infinitely many times

$$\int_{\mathcal{S}}\int_{\mathcal{Z}} \iota(s_0, z_0)\left(G^{\mu_{\boldsymbol{\theta}}}(s_0, z_0) + \iota(s_0, z_0)\int_{\mathcal{S}}\int_{\mathcal{Z}}\gamma P^{\mu_{\boldsymbol{\theta}}}(s_1, z_1|s_0, z_0)\nabla_{\boldsymbol{\theta}}V^{\mu_{\boldsymbol{\theta}}}(s_1, z_1)ds_1 dz_1\right)ds_0 dz_0$$

$$= \int_{\mathcal{S}}\int_{\mathcal{Z}}\mathrm{Tr}_0^{\mu_{\boldsymbol{\theta}}}(s_0, z_0)G^{\mu_{\boldsymbol{\theta}}}(s_0, z_0)ds_0 dz_0 + \int_{\mathcal{S}}\int_{\mathcal{Z}}\int_{\mathcal{S}}\int_{\mathcal{Z}}\gamma\iota(s_0, z_0)P^{\mu_{\boldsymbol{\theta}}}(s_1, z_1|s_0, z_0)\cdot$$

$$\cdot\left(G^{\mu_{\boldsymbol{\theta}}}(s_1, z_1) + \int_{\mathcal{S}}\int_{\mathcal{Z}}\gamma P^{\mu_{\boldsymbol{\theta}}}(s_2, z_2|s_1, z_1)\nabla_{\boldsymbol{\theta}}V^{\mu_{\boldsymbol{\theta}}}(s_2, z_2)ds_2 dz_2\right)ds_1 dz_1$$

$$= \int_{\mathcal{S}}\int_{\mathcal{Z}}\mathrm{Tr}_0^{\mu_{\boldsymbol{\theta}}}(s_0, z_0)G^{\mu_{\boldsymbol{\theta}}}(s_0, z_0)ds_0 dz_0 + \int_{\mathcal{S}}\int_{\mathcal{Z}}\gamma\mathrm{Tr}_1^{\mu_{\boldsymbol{\theta}}}(s_1, z_1)G^{\mu_{\boldsymbol{\theta}}}(s_1, z_1)ds_1 dz_1$$

$$+ \int_{\mathcal{S}^3 \times \mathcal{Z}^3}\gamma^2\iota(s_0, z_0)P^{\mu_{\boldsymbol{\theta}}}(s_1, z_1|s_0, z_0)P^{\mu_{\boldsymbol{\theta}}}(s_2, z_2|s_1, z_1)\nabla_{\boldsymbol{\theta}}V^{\mu_{\boldsymbol{\theta}}}(s_2, z_2)ds_2 dz_2 ds_1 dz_1 ds_0 dz_0$$

$$= \sum_{t=0}^{\infty}\int_{\mathcal{S}}\int_{\mathcal{Z}}\gamma^t \mathrm{Tr}_t^{\mu_{\boldsymbol{\theta}}}(s_t, z_t)G^{\mu_{\boldsymbol{\theta}}}(s_t, z_t)ds_t dz_t. \tag{16}$$

Now, we notice that the occupancy measure is defined as

$$\rho^{\mu_{\boldsymbol{\theta}}}(s, z) = \sum_{t=0}^{\infty}\gamma^t \mathrm{Tr}_t^{\mu_{\boldsymbol{\theta}}}(s_t, z_t).$$

Therefore, by exchanging the order of series and integration we obtain:

$$\nabla_{\boldsymbol{\theta}}\mathcal{J}(\mu_{\boldsymbol{\theta}}) = \int_{\mathcal{S}}\int_{\mathcal{Z}}\sum_{t=0}^{\infty}\gamma^t \mathrm{Tr}_t^{\mu_{\boldsymbol{\theta}}}(s, z)G^{\mu_{\boldsymbol{\theta}}}(s, z)ds dz$$

$$= \int_{\mathcal{S}}\int_{\mathcal{Z}}\rho^{\mu_{\boldsymbol{\theta}}}(s, z)\left(\nabla_{\boldsymbol{\theta}}\mu_{\boldsymbol{\theta}}^a(s, z)\nabla_a Q^{\mu_{\boldsymbol{\theta}}}(s, z, a, \mu_{\boldsymbol{\theta}}^z(s, z))|_{a=\mu_{\boldsymbol{\theta}}^a(s, z)}\right.$$

$$\left. +\nabla_{\boldsymbol{\theta}}\mu_{\boldsymbol{\theta}}^z(s, z)\,\nabla_{z'}Q^{\mu_{\boldsymbol{\theta}}}(s, z, \mu_{\boldsymbol{\theta}}^a(s, z), z')|_{z'=\mu_{\boldsymbol{\theta}}^z(s, z)}\right),$$

which concludes the proof. $\qquad\square$

## A.6 Variance Analysis of Policy Gradient Estimates

In the following section, we define the variance of a random vector $A = (A_1, \ldots, A_l)^\top$ as in Papini et al. (2022); Zhao et al. (2011)

$$\text{Var}[A] = \text{tr}\left(\mathbb{E}\left[(A - \mathbb{E}[A])(A - \mathbb{E}[A])^\top\right]\right) \tag{17}$$

$$= \sum_{m=1}^{l} \mathbb{E}\left[(A_m - \mathbb{E}[A_m])^2\right] \tag{18}$$

Using this definition, we can compute the variance of the policy gradient estimators for stateful policies using BPTT and S2PG.

### A.6.1 Backpropagation Through Time

*Proof.* We consider a policy of the following form

$$\nu(a_t|h_t, \Theta) = \mathcal{N}(a_t|\mu_\theta(h_t), \Sigma) = \frac{1}{\sqrt{2\pi^d|\Sigma|}} \exp\left(-\frac{1}{2}(a_t - \mu_\theta(h_t))^\top \Sigma^{-1}(a_t - \mu_\theta(h_t))\right), \tag{19}$$

where $\Theta = \{\theta, \Sigma\}$. For a fixed covariance $\Sigma$, a REINFORCE-style gradient estimator for this policy depends on the following gradient

$$f(\tau) = \sum_{t=0}^{T-1} \nabla_\theta \log \pi(a_t|h_t, \Theta) = \sum_{t=0}^{T-1} \nabla_\theta \mu_\theta(h_t) \nabla_{m_t} \log \mathcal{N}(a_t|m_t, \Sigma)|_{m_t = \mu_\theta(h_t)}$$

$$= \sum_{t=0}^{T-1} \nabla_\theta \mu_\theta(h_t) \left(\Sigma^{-1}(a_t - m_t)\right)|_{m_t = \mu_\theta(h_t)}.$$

Given the total discounted return of a trajectory

$$G(\tau) = \sum_{t=0}^{T-1} \gamma^{t-1} r(s_t, a_t, s_{t+1}),$$

the variance of the empirical gradient approximator is given by

$$\text{Var}\left[\nabla_\theta \hat{J}(\Theta)\right] = \frac{1}{N} \text{Var}\left[G(\tau)f(\tau)\right], \tag{20}$$

where $N$ is the number of trajectories used for the empirical gradient estimator. Therefore, we can just focus on $\text{Var}\left[G(\tau)f(\tau)\right]$. Using equation 18, we can define an upper bound on $\text{Var}\left[G(\tau)f(\tau)\right]$

$$\text{Var}\left[G(\tau)f(\tau)\right] \leq \sum_{i=1}^{l} \mathbb{E}\left[(Gf_i)^2\right]$$

$$= \mathbb{E}\left[G^2 f^\top f\right]$$

$$= \int_\tau p(\tau) \left(\sum_{t=0}^{T-1} \gamma^{t-1} r(s_t, a_t, s_{t+1})\right)^2$$

$$\cdot \left(\sum_{t=0}^{T-1} \nabla_\theta \mu_\theta(h_t) \left(\Sigma^{-1}(a_t - m_t)\right)|_{m_t = \mu_\theta(h_t)}\right)^\top$$

$$\cdot \left(\sum_{t=0}^{T-1} \nabla_\theta \mu_\theta(h_t) \left(\Sigma^{-1}(a_t - m_t)\right)|_{m_t = \mu_\theta(h_t)}\right).$$

Let $\xi_t = \Sigma^{-1}(a_t - m_t)$ where $a_t \sim \mathcal{N}(m_t, \Sigma)$, then $\xi_t(m_t)$ are random variables drawn from a Gaussian distribution $\mathcal{N}(0, \Sigma^{-1})$. Hence, we define $p(\xi_t) = \mathcal{N}(0, \Sigma^{-1})$ and treat $\xi_t$ as an independent random variable sampled from $p(\xi_t)$. For compactness of notation, we define $\xi = [\xi_0, \ldots, \xi_T]^\top$, where $p(\xi) = \prod_{t=0}^{T} p(\xi_t)$. In a similar way we define

$p(\tau|\xi) = \iota(s_0) \prod_{t=0}^{T} P(s_{t+1}|s_t, \xi_t)$. Using this notation, we can write

$$
\begin{aligned}
\text{Var}\left[G(\tau)f(\tau)\right] &\leq \int_{\tau,\xi} p(\xi)\, p(\tau|\xi) \left(\sum_{t=0}^{T-1} \gamma^{t-1} r(s_t, a_t, s_{t+1})\right)^2 \left(\sum_{t=0}^{T-1} \nabla_\theta \mu_\theta(h_t)\, \xi_t\right)^\top \left(\sum_{t=0}^{T-1} \nabla_\theta \mu_\theta(h_t)\, \xi_t\right) \\
&\leq \frac{R^2(1-\gamma^T)^2}{(1-\gamma)^2} \; \mathbb{E}_{\tau,\xi}\left[\sum_{t=0}^{T-1}\sum_{t'=1}^{T} \xi_t^\top \nabla_\theta \mu_\theta(h_t)^\top \nabla_\theta \mu_\theta(h_{t'})\, \xi_t\right] \\
&= \frac{R^2(1-\gamma^T)^2}{(1-\gamma)^2} \left(\mathbb{E}_{\tau,\xi}\left[\sum_{t=0}^{T-1} \xi_t^\top \nabla_\theta \mu_\theta(h_t)^\top \nabla_\theta \mu_\theta(h_t)\, \xi_t\right]\right. \\
&\qquad\qquad\left. + \mathbb{E}_{\tau,\xi}\left[\sum_{t,t'=0, t\neq t'}^{T-1} \xi_t^\top \nabla_\theta \mu_\theta(h_t)^\top\, \nabla_\theta \mu_\theta(h_{t'})\, \xi_t\right]\right).
\end{aligned}
\tag{21}
$$

Now, we take a look at the last term. Without loss of generality, we will assume $t' > t$. As $\xi_t$ is a random variable drawn from a Gaussian distribution with zero mean, and as $h_{t'}$ and $h_t$ are independent from $\xi_{t'}$, we can write

$$
\begin{aligned}
\mathbb{E}_{\tau,\xi}&\left[\sum_{t,t'=0, t\neq t'}^{T-1} \xi_t^\top \nabla_\theta \mu_\theta(h_t)^\top\, \nabla_\theta \mu_\theta(h_{t'})\xi_{t'}\right] \\
&= \sum_{t,t'=0, t\neq t'}^{T-1} \mathbb{E}_{\tau,\xi}\left[\xi_t^\top \nabla_\theta \mu_\theta(h_t)^\top\, \nabla_\theta \mu_\theta(h_{t'})\xi_{t'}\right] \\
&= \sum_{t,t'=0, t\neq t'}^{T-1} \mathbb{E}_{\tau,\xi}\left[\xi_t^\top \nabla_\theta \mu_\theta(h_t)^\top \nabla_\theta \mu_\theta(h_{t'})\right] \mathbb{E}_{\xi_{t'}}\left[\xi_{t'}\right] \\
&= 0\,.
\end{aligned}
$$

where in the last line we used the fact that $\mathbb{E}_{\xi_{t'}}\left[\xi_{t'}\right] = \mathbf{0}$. The same result holds for $t > t'$, following a similar derivation.

Continuing from equation 21, let $\Xi_t = \nabla_\theta \mu_\theta(h_t)^\top \nabla_\theta \mu_\theta(h_t)$. Noticing that $\Xi_t$ is a square positive semi-definite matrix, we can write

$$
\begin{aligned}
\frac{R^2(1-\gamma^T)^2}{(1-\gamma)^2}\; &\mathbb{E}_{\tau,\xi}\left[\sum_{t=0}^{T-1} \xi_t^\top \nabla_\theta \mu_\theta(h_t)^\top \nabla_\theta \mu_\theta(h_t)\xi_t\right] \\
&= \frac{R^2(1-\gamma^T)^2}{(1-\gamma)^2} \sum_{t=0}^{T-1} \mathbb{E}_{\tau,\xi}\left[\mathbb{E}_{\xi_t}\left[\xi_t^\top \nabla_\theta \mu_\theta(h_t)^\top \nabla_\theta \mu_\theta(h_t)\xi_t\right]\right] \\
&= \frac{R^2(1-\gamma^T)^2}{(1-\gamma)^2} \sum_{t=0}^{T-1} \mathbb{E}_{\tau,\xi}\left[\text{tr}\left(\nabla_\theta \mu_\theta(h_t)^\top \nabla_\theta \mu_\theta(h_t)\, \Sigma^{-1}\right)\right] \\
&\leq \frac{R^2(1-\gamma^T)^2}{(1-\gamma)^2} \sum_{t=0}^{T-1} \mathbb{E}_{\tau,\xi}\left[\left\|\nabla_\theta \mu_\theta(h_t)^\top\right\|_F \cdot \left\|\nabla_\theta \mu_\theta(h_t)\right\|_F \cdot \left\|\Sigma^{-1}\right\|_F\right] \\
&\leq \frac{R^2(1-\gamma^T)^2}{(1-\gamma)^2} \sum_{t=0}^{T-1} \mathbb{E}_{\tau,\xi}\left[\left\|\nabla_\theta \mu_\theta(h_t)\right\|_F^2 \cdot \left\|\Sigma^{-1}\right\|_F\right]\,,
\end{aligned}
$$

where $\|.\|_F$ is the Frobenius norm. When $\mu(h_t)$ is implemented as a recursive function $\mu(h_t) = f_\theta(s_t, z_{t+1})$ where $z_{t+1} = \eta_\theta(s_t, z_t)$ the gradient is given by

$$
\nabla_\theta \mu_\theta(h_t) = \frac{\partial}{\partial \theta} f_\theta(s_t, z_t) + \left(\sum_{i=0}^{t-1} \frac{\partial}{\partial \theta}\eta_\theta(s_i, z_i) \prod_{j=i+1}^{t} \frac{\partial}{\partial z_j}\eta_\theta(s_j, z_j)\right) \frac{\partial}{\partial z_t} f_\theta(s_t, z_t)\,.
\tag{22}
$$

Using this gradient and using the Assumptions 3.2 and 3.3, we bound the norm of the gradient of $\mu_\theta(h_t)$ as follows

$$
\begin{aligned}
\left\|\nabla_\theta \mu_\theta(h_t)\right\|_{\mathrm{F}} &\leq \left\|\frac{\partial}{\partial\theta}f_\theta(s_t, z_t)\right\|_{\mathrm{F}} + \left(\sum_{i=0}^{t-1}\left\|\frac{\partial}{\partial\theta}\eta_\theta(s_i, z_i)\right\|_{\mathrm{F}} \prod_{j=i+1}^{t}\left\|\frac{\partial}{\partial z_j}\eta_\theta(s_j, z_j)\right\|_{\mathrm{F}}\right)\left\|\frac{\partial}{\partial z_t}f_\theta(s_t, z_t)\right\|_{\mathrm{F}} \\
&\leq F + \left(\sum_{i=0}^{t-1} H \prod_{j=i+1}^{t} Z\right) K \\
&= F + H \left(\sum_{i=0}^{t-1} Z^{t-i-1}\right) K \,.
\end{aligned}
$$

Hence, the final bound is given by

$$
\begin{aligned}
\mathrm{Var}\left[\nabla_\theta \hat{J}_{\mathrm{BPTT}}(\Theta)\right] &\leq \frac{R^2\left\|\Sigma^{-1}\right\|_{\mathrm{F}}(1-\gamma^T)^2}{N(1-\gamma)^2} \sum_{t=0}^{T-1}\left(F + H\left(\sum_{i=0}^{t-1} Z^{t-i-1}\right) K\right)^2 \\
&= \frac{R^2\left\|\Sigma^{-1}\right\|_{\mathrm{F}}(1-\gamma^T)^2}{N(1-\gamma)^2}\left(TF^2 + 2FHK\tilde{Z} + H^2K^2\bar{Z}^2\right),
\end{aligned}
$$

where we define $\tilde{Z} = \sum_{t=0}^{T-1}\sum_{i=0}^{t-1} Z^{t-i-1}$ and $\bar{Z} = \sum_{t=0}^{T-1}\left(\sum_{i=0}^{t-1} Z^{t-i-1}\right)^2$ for brevity, concluding the proof. $\qquad\square$

We can further analyze the properties of the BPTT estimator with the following lemma:

**Lemma A.2.** *For $Z < 1$ the two constants $\tilde{Z}$ and $\bar{Z}$ grow linearly with $T$, while for $Z > 1$ the two constants grow exponentially with $T$.*

*Proof.* The series expressing $\bar{Z}$ can be written as

$$
\begin{aligned}
\hat{Z} = \sum_{t=0}^{T-1}\left(\sum_{i=0}^{t-1} Z^{t-i-1}\right)^2 &= \sum_{t=0}^{T-1}\left(\sum_{k=0}^{t-1} Z^k\right)^2 = \sum_{t=0}^{T-1}\frac{(1-Z^t)^2}{(1-Z)^2} \\
&= \frac{1}{(1-Z)^2}\left(T + \sum_{t=0}^{T-1} Z^{2t} - 2\sum_{t=0}^{T-1} Z^t\right) \\
&= \frac{T}{(1-Z)^2} + \frac{1-Z^{2T}}{1-Z^2} - 2\frac{1-Z^T}{1-Z} \\
&= \frac{T}{(1-Z)^2} + \frac{1}{(1-Z)^2}\left(\frac{1-Z^{2T}-2(1-Z^T)(1+Z)}{(1-Z)(1+Z)}\right) \\
&= \frac{T}{(1-Z)^2} + \frac{(1-Z^T)(1+Z^T) - 2(1-Z^T)(1+Z)}{(1-Z)^3(1+Z)} \\
&= \frac{T}{(1-Z)^2} + \frac{(1-Z^T)(Z^T - 2Z - 1)}{(1-Z)^3(1+Z)} \\
&= \frac{T}{(1-Z)^2} + \frac{(Z^T - 1)(Z^T - 2Z - 1)}{(Z-1)^3(1+Z)}
\end{aligned}
\tag{23}
$$

From equation 23 we see that $\bar{Z}$ is composed of two terms, where the first is $\mathcal{O}(T)$ while the second one is $\mathcal{O}(Z^{2T})$. The series expressing $\tilde{Z}$ can be written as

$$\tilde{Z} = \sum_{t=0}^{T-1}\sum_{i=0}^{t-1} Z^{t-i-1} = \sum_{t=0}^{T-1}\sum_{k=0}^{t-1} Z^k = \sum_{t=0}^{T-1} \frac{1 - Z^t}{1 - Z}$$

$$= \frac{1}{1 - Z}\left(T - \sum_{t=0}^{T-1} Z^t\right) = \frac{T}{1 - Z} + \frac{Z^T - 1}{(1 - Z)^2} \tag{24}$$

From equation 24 we see that $\tilde{Z}$ is composed of two terms, where the first is $\mathcal{O}(T)$ while the second one is $\mathcal{O}(Z^T)$. These observations conclude the proof. $\square$

### A.6.2 STOCHASTIC STATEFUL POLICIES

*Proof.* For stochastic stateful policies, we consider similar policies of the form

$$\pi(a_t, z_{t+1}|s_t, z_t, \tilde{\Theta}) = \mathcal{N}(\vartheta_\theta(s_t, z_{t+1}), \Gamma) \tag{25}$$

$$= \frac{1}{\sqrt{2\pi^d|\Gamma|}} \exp\left(-\frac{1}{2}(a_t - \vartheta_\theta(s_t, z_t))^\top \Gamma^{-1}(a_t - \vartheta_\theta(s_t, z_t))\right),$$

where $\tilde{\Theta} = \{\theta, \Gamma\}$, where $\vartheta_\theta$ is the mean vector containing the means corresponding to an action and a hidden state such that $\vartheta_\theta(s_t, z_t) = \left[\mu_\theta^a(s_t, z_t)^\top, \mu_\theta^z(s_t, z_t)^\top\right]^\top$, and $\Gamma$ is a covariance matrix containing the covariances corresponding to an action and a hidden state such that $\Gamma = \begin{bmatrix} \Sigma & 0 \\ 0 & \Upsilon \end{bmatrix}$.

Then, we can write the gradient with respect to a trajectory as follows

$$\tilde{f}(\bar{\tau}) = \sum_{t=0}^{T-1} \nabla_\theta \log \pi(a_t, z_{t+1}|s_t, z_t, \tilde{\Theta}) = \sum_{t=0}^{T-1} \nabla_\theta \left(\log \pi(a_t|s_t, z_t, \tilde{\Theta}) + \log \pi(z_{t+1}|s_t, z_t, \tilde{\Theta})\right)$$

$$= \sum_{t=0}^{T-1} \nabla_\theta \mu_\theta^a(s_t, z_t) \nabla_{m_t} \log \mathcal{N}(m_t, \Sigma)|_{m_t = \mu_\theta(s_t, z_t)}$$

$$+ \nabla_\theta \mu_\theta^z(s_t, z_t) \nabla_{b_t} \log \mathcal{N}(b_t, \Upsilon)|_{b_t = \mu_\theta^z(s_t, z_t)}$$

$$= \sum_{t=0}^{T-1} \nabla_\theta \mu_\theta^a(s_t, z_t)\left(\Sigma^{-1}(a_t - m_t)\right)|_{m_t = \mu_\theta(s_t, z_t)}$$

$$+ \nabla_\theta \mu_\theta^z(s_t, z_t)\left(\Upsilon^{-1}(z_{t+1} - b_t)\right)|_{b_t = \mu_\theta^z(s_t, z_t)},$$

where we have used the fact that $a_t$ and $z_{t+1}$ are independent for decomposing the gradient into a part corresponding to the mean of the Gaussian used for sampling action and a part corresponding to the mean of the Gaussian used to sample the next hidden state.

Let $\tilde{\xi}_t = \Upsilon^{-1}(z_{t+1} - b_t)$ where $z_{t+1} \sim \mathcal{N}(b_t, \Upsilon)$, then $\tilde{\xi}_t(b_t)$ are random variables drawn from a Gaussian distribution $\mathcal{N}(0, \Upsilon^{-1})$. Hence, we define $p(\tilde{\xi}_t) = \mathcal{N}(0, \Upsilon^{-1})$ and treat $\tilde{\xi}_t$ as an independent random variable sampled from $p(\tilde{\xi}_t)$. Then we can use equation 18 again to define an upper

bound on the variance such that

$$
\begin{aligned}
\mathrm{Var}\left[G(\bar\tau)\tilde f(\bar\tau)\right] &\leq \int_{\bar\tau,\xi} p(\bar\tau)\,p(\xi)\,p(\tilde\xi)\left(\sum_{t=0}^{T-1}\gamma^{t-1}r(s_t,a_t,s_{t+1})\right)^2 \\
&\quad\cdot\left(\sum_{t=0}^{T-1}\nabla_\theta\mu_\theta^a(s_t,z_t)\,\xi_t+\nabla_\theta\mu_\theta^z(s_t,z_t)\tilde\xi_t\right)^\top \\
&\quad\cdot\left(\sum_{t=0}^{T-1}\nabla_\theta\mu_\theta^a(s_t,z_t)\,\xi_t+\nabla_\theta\mu_\theta^z(s_t,z_t)\tilde\xi_t\right) \\
&\leq\frac{R^2(1-\gamma^T)^2}{(1-\gamma)^2}\mathop{\mathbb{E}}_{\bar\tau,\xi,\tilde\xi}\left[\sum_{t=0}^{T-1}\sum_{t'=1}^{T}\left(\nabla_\theta\mu_\theta^a(s_t,z_t)\,\xi_t+\nabla_\theta\mu_\theta^z(s_t,z_t)\tilde\xi_t\right)^\top\right. \\
&\quad\left.\cdot\left(\nabla_\theta\mu_\theta^a(s_{t'},z_{t'})\,\xi_{t'}+\nabla_\theta\mu_\theta^z(s_{t'},z_{t'})\tilde\xi_{t'}\right)\right] \\
&=\frac{R^2(1-\gamma^T)^2}{(1-\gamma)^2}\mathop{\mathbb{E}}_{\bar\tau,\xi,\tilde\xi}\left[\sum_{t=0}^{T-1}\left(\nabla_\theta\mu_\theta^a(s_t,z_t)\,\xi_t+\nabla_\theta\mu_\theta^z(s_t,z_t)\tilde\xi_t\right)^\top\right. \\
&\quad\left.\cdot\left(\nabla_\theta\mu_\theta^a(s_t,z_t)\,\xi_t+\nabla_\theta\mu_\theta^z(s_t,z_t)\tilde\xi_t\right)\right] \\
&\quad\cdot\mathop{\mathbb{E}}_{\bar\tau,\xi,\tilde\xi}\left[\sum_{t,t'=0,t\neq t'}^{T-1}\left(\nabla_\theta\mu_\theta^a(s_t,z_t)\,\xi_t+\nabla_\theta\mu_\theta^z(s_t,z_t)\tilde\xi_t\right)^\top\right. \\
&\quad\left.\cdot\left(\nabla_\theta\mu_\theta^a(s_{t'},z_{t'})\,\xi_{t'}+\nabla_\theta\mu_\theta^z(s_{t'},z_{t'})\tilde\xi_{t'}\right)\right].
\end{aligned}
$$

Again, the last term can be dropped due to

$$
\begin{aligned}
&\mathop{\mathbb{E}}_{\bar\tau,\xi,\tilde\xi}\left[\sum_{t,t'=0,t\neq t'}^{T-1}\left(\nabla_\theta\mu_\theta^a(s_t,z_t)\,\xi_t+\nabla_\theta\mu_\theta^z(s_t,z_t)\tilde\xi_t\right)^\top\left(\nabla_\theta\mu_\theta^a(s_{t'},z_{t'})\,\xi_{t'}+\nabla_\theta\mu_\theta^z(s_{t'},z_{t'})\tilde\xi_{t'}\right)\right] \\
&=\sum_{t,t'=0,t\neq t'}^{T-1}\mathop{\mathbb{E}}_{\bar\tau,\xi,\tilde\xi}\left[\left(\nabla_\theta\mu_\theta^a(s_t,z_t)\,\xi_t+\nabla_\theta\mu_\theta^z(s_t,z_t)\tilde\xi_t\right)^\top\left(\nabla_\theta\mu_\theta^a(s_{t'},z_{t'})\,\xi_{t'}+\nabla_\theta\mu_\theta^z(s_{t'},z_{t'})\tilde\xi_{t'}\right)\right] \\
&=\sum_{t,t'=0,t\neq t'}^{T-1}\mathop{\mathbb{E}}_{\bar\tau,\xi,\tilde\xi}\left[\left(\nabla_\theta\mu_\theta^a(s_t,z_t)\,\xi_t+\nabla_\theta\mu_\theta^z(s_t,z_t)\tilde\xi_t\right)^\top\right] \\
&\quad\cdot\mathop{\mathbb{E}}_{\bar\tau,\xi,\tilde\xi}\left[\left(\nabla_\theta\mu_\theta^a(s_{t'},z_{t'})\,\xi_{t'}+\nabla_\theta\mu_\theta^z(s_{t'},z_{t'})\tilde\xi_{t'}\right)\right] \\
&=\sum_{t,t'=0,t\neq t'}^{T-1}\Big(\underbrace{\mathop{\mathbb{E}}_{\bar\tau,\xi,}\left[\nabla_\theta\mu_\theta^a(s_t,z_t)^\top\xi_t^\top\right]}_{=0}+\underbrace{\mathop{\mathbb{E}}_{\bar\tau,\tilde\xi}\left[\nabla_\theta\mu_\theta^z(s_t,z_t)^\top\tilde\xi_t^\top\right]}_{=0}\Big) \\
&\quad\cdot\Big(\underbrace{\mathop{\mathbb{E}}_{\bar\tau,\xi}\left[\nabla_\theta\mu_\theta^a(s_{t'},z_{t'})\,\xi_{t'}\right]}_{=0}+\underbrace{\mathop{\mathbb{E}}_{\bar\tau,\tilde\xi}\left[\nabla_\theta\mu_\theta^z(s_{t'},z_{t'})\tilde\xi_{t'}\right]}_{=0}\Big)=0.
\end{aligned}
$$

Noticing that $\nabla_\theta\mu_\theta^a(s_t,z_t)^\top\nabla_\theta\mu_\theta^a(s_t,z_t)$ and $\nabla_\theta\mu_\theta^z(s_t,z_t)^\top\nabla_\theta\mu_\theta^z(s_t,z_t)$ are square positive semi-definite matrices, allows us to write

$$\frac{R^2(1-\gamma^T)^2}{(1-\gamma)^2} \mathop{\mathbb{E}}_{\bar{\tau},\xi,\tilde{\xi}} \left[ \sum_{t=0}^{T-1} \left( \nabla_\theta \mu_\theta^a(s_t, z_t)\, \xi_t + \nabla_\theta \mu_\theta^z(s_t, z_t)\tilde{\xi}_t \right)^\top \left( \nabla_\theta \mu_\theta^a(s_t, z_t)\, \xi_t + \nabla_\theta \tilde{\mu}_\theta(s_t, z_t)\tilde{\xi}_t \right) \right]$$

$$= \frac{R^2(1-\gamma^T)^2}{(1-\gamma)^2} \sum_{t=0}^{T-1} \mathop{\mathbb{E}}_{\bar{\tau},\xi,\tilde{\xi}} \left[ \left( \nabla_\theta \mu_\theta^a(s_t, z_t)\, \xi_t + \nabla_\theta \mu_\theta^z(s_t, z_t)\tilde{\xi}_t \right)^\top \left( \nabla_\theta \mu_\theta^a(s_t, z_t)\, \xi_t + \nabla_\theta \tilde{\mu}_\theta(s_t, z_t)\tilde{\xi}_t \right) \right]$$

$$= \frac{R^2(1-\gamma^T)^2}{(1-\gamma)^2}$$

$$\cdot \left( \sum_{t=0}^{T-1} \mathop{\mathbb{E}}_{\bar{\tau},\xi} \left[ \left( \nabla_\theta \mu_\theta^a(s_t, z_t)\, \xi_t \right)^\top \left( \nabla_\theta \mu_\theta^a(s_t, z_t)\, \xi_t \right) \right] + \mathop{\mathbb{E}}_{\bar{\tau},\tilde{\xi}} \left[ \left( \nabla_\theta \mu_\theta^z(s_t, z_t)\tilde{\xi}_t \right)^\top \left( \nabla_\theta \mu_\theta^z(s_t, z_t)\tilde{\xi}_t \right) \right] \right.$$

$$\left. + \underbrace{\mathop{\mathbb{E}}_{\bar{\tau},\xi,\tilde{\xi}} \left[ \left( \nabla_\theta \mu_\theta^a(s_t, z_t)\, \xi_t \right)^\top \left( \nabla_\theta \mu_\theta^z(s_t, z_t)\tilde{\xi}_t \right) \right]}_{=0} + \underbrace{\mathop{\mathbb{E}}_{\bar{\tau},\xi,\tilde{\xi}} \left[ \left( \nabla_\theta \mu_\theta^z(s_t, z_t)\tilde{\xi}_t \right)^\top \left( \nabla_\theta \mu_\theta^a(s_t, z_t)\, \xi_t \right) \right]}_{=0} \right)$$

$$= \frac{R^2(1-\gamma^T)^2}{(1-\gamma)^2} \left( \sum_{t=0}^{T-1} \mathop{\mathbb{E}}_{\bar{\tau}} \left[ \mathrm{tr}\left( \nabla_\theta \mu_\theta^a(s_t, z_t)^\top \nabla_\theta \mu_\theta^a(s_t, z_t)\, \Sigma^{-1} \right) \right] \right.$$

$$\left. + \mathop{\mathbb{E}}_{\bar{\tau}} \left[ \mathrm{tr}\left( \nabla_\theta \mu_\theta^z(s_t, z_t)^\top \nabla_\theta \mu_\theta^z(s_t, z_t)\, \Upsilon^{-1} \right) \right] \right)$$

$$\leq \frac{R^2(1-\gamma^T)^2}{(1-\gamma)^2} \left( \sum_{t=0}^{T-1} \mathop{\mathbb{E}}_{\bar{\tau}} \left[ \left\| \nabla_\theta \mu_\theta^a(s_t, z_t) \right\|_{\mathrm{F}}^2 \cdot \left\| \Sigma^{-1} \right\|_{\mathrm{F}} \right] + \mathop{\mathbb{E}}_{\bar{\tau}} \left[ \left\| \nabla_\theta \mu_\theta^z(s_t, z_t) \right\|_{\mathrm{F}}^2 \cdot \left\| \Upsilon^{-1} \right\|_{\mathrm{F}} \right] \right).$$

To align with the notation from the previous proof, we define $\mu_\theta^a(s_t, z_t) = f_\theta(s_t, z_t)$ and $\mu_\theta^z(s_t, z_t) = \eta_\theta(s_t, z_t)$ and get

$$\frac{R^2(1-\gamma^T)^2}{(1-\gamma)^2} \left( \sum_{t=0}^{T-1} \mathop{\mathbb{E}}_{\bar{\tau}} \left[ \left\| \nabla_{\boldsymbol\theta} f_{\boldsymbol\theta}(s_t, z_t) \right\|_{\mathrm{F}}^2 \cdot \left\| \Sigma^{-1} \right\|_{\mathrm{F}} \right] + \mathop{\mathbb{E}}_{\bar{\tau}} \left[ \left\| \nabla_{\boldsymbol\theta} \nu_{\boldsymbol\theta}(s_t, z_t) \right\|_{\mathrm{F}}^2 \cdot \left\| \Upsilon^{-1} \right\|_{\mathrm{F}} \right] \right)$$

$$\leq \frac{R^2(1-\gamma^T)^2}{(1-\gamma)^2} \left( \sum_{t=0}^{T-1} F^2 \cdot \left\| \Sigma^{-1} \right\|_{\mathrm{F}} + H^2 \cdot \left\| \Upsilon^{-1} \right\|_{\mathrm{F}} \right)$$

such that the final bound on the variance of the S2PG is given by

$$\mathrm{Var}\left[ \nabla_\theta \hat{J}_{\mathrm{S2PG}}(\Theta) \right] \leq \frac{R^2 \left\| \Sigma^{-1} \right\|_{\mathrm{F}} (1-\gamma^T)^2}{N(1-\gamma)^2} \left( \sum_{t=0}^{T-1} F^2 + H^2 \cdot \frac{\left\| \Upsilon^{-1} \right\|_{\mathrm{F}}}{\left\| \Sigma^{-1} \right\|_{\mathrm{F}}} \right),$$

concluding the proof. $\qquad\qquad\square$

**Lemma A.3.** *Given a policy, similarily to equation 25, $\pi(a_t, z_{t+1}|s_t, z_t, \tilde{\Theta}) = \mathcal{N}(\vartheta_\theta(s_t, z_{t+1}), \Gamma)$ with $\Gamma = \begin{bmatrix} \Sigma & 0 \\ 0 & \Upsilon \end{bmatrix}$ and limiting the covariance matrices $\Sigma$ and $\Upsilon$ to be diagonal, then the bound from Theorem 3.5 can be replaced by a tighter bound*

$$\mathrm{Var}\left[ \nabla_\theta \hat{J}_{S2PG}(\Theta) \right] \leq \frac{R^2 \, \mathrm{tr}(\Sigma^{-1})(1-\gamma^T)^2 T}{N(1-\gamma)^2} \left( F_d^2 + H_d^2 \frac{\mathrm{tr}(\Upsilon^{-1})}{\mathrm{tr}(\Sigma^{-1})} \right).$$

*Proof.* Indeed, when $B$ is diagonal it is easy to show that

$$\mathrm{tr}(AB) = \mathrm{diag}(A)^\top \mathrm{diag}(B). \tag{26}$$

Let $\mathrm{diag}(\Sigma) = \boldsymbol\sigma = [\sigma_0, \ldots, \sigma_{|a|}]^\top$, and $\mathrm{diag}(\Upsilon) = \boldsymbol{v} = [v_0, \ldots, v_{|z|}]^\top$. Using equation 26 we can write

$$\mathrm{tr}\left( \nabla_\theta \mu_\theta^a(s_t, z_t)^\top \nabla_\theta \mu_\theta^a(s_t, z_t)\, \Sigma^{-1} \right) = \mathrm{diag}\left( \nabla_\theta \mu_\theta^a(s_t, z_t)^\top \nabla_\theta \mu_\theta^a(s_t, z_t) \right)^\top \mathrm{diag}\left( \Sigma^{-1} \right)$$

$$\mathrm{tr}\left( \nabla_\theta \mu_\theta^z(s_t, z_t)^\top \nabla_\theta \mu_\theta^z(s_t, z_t)\, \Upsilon^{-1} \right) = \mathrm{diag}\left( \nabla_\theta \mu_\theta^z(s_t, z_t)^\top \nabla_\theta \mu_\theta^z(s_t, z_t) \right)^\top \mathrm{diag}\left( \Upsilon^{-1} \right).$$

Let $X^{[i]}$ be the $i$-th row of the matrix $X$. We observe that

$$
\mathrm{diag}\left(\nabla_\theta \mu_\theta^a(s_t, z_t)^\top \nabla_\theta \mu_\theta^a(s_t, z_t)\right) =
\begin{bmatrix}
\nabla_\theta(\mu_\theta^a(s_t, z_t)^{[0]})^\top \nabla_\theta \mu_\theta^a(s_t, z_t)^{[0]} \\
\cdots \\
\nabla_\theta(\mu_\theta^a(s_t, z_t)^{[|a|-1]})^\top \nabla_\theta \mu_\theta^a(s_t, z_t)^{[|a|-1]}
\end{bmatrix}
$$

$$
=
\begin{bmatrix}
\|\nabla_\theta \mu_\theta^a(s_t, z_t)^{[0]}\|_2^2 \\
\cdots \\
\|\nabla_\theta \mu_\theta^a(s_t, z_t)^{[|a|-1]}\|_2^2
\end{bmatrix}.
$$

The same observation can be done by exchanging $\mu_\theta^a$ with $\mu_\theta^z$. Notice that the resulting vector is a vector of the norms of the gradient of the mean functions w.r.t. a specific action (or internal state component).

If we assume the existence of two positive constants $F_d$ and $H_d$ such that $\|\nabla_\theta \mu_\theta^a(s_t, z_t)^{[i]}\|_2^2 \leq F_d^2$ and $\|\nabla_\theta \mu_\theta^z(s_t, z_t)^{[i]}\|_2^2 \leq H_d^2$, $\forall i, s_t, z_t$ then we can write

$$
\mathrm{tr}\left(\nabla_\theta \mu_\theta^a(s_t, z_t)^\top \nabla_\theta \mu_\theta^a(s_t, z_t) \Sigma^{-1}\right) \leq F_d^2 \cdot \left(\mathbf{1}^\top \mathrm{diag}(\Sigma^{-1})\right)
$$

$$
= F_d^2 \sum_{i=0}^{|a|-1} \frac{1}{\sigma_i} = F_d^2 \, \mathrm{tr}(\Sigma^{-1})
$$

$$
\mathrm{tr}\left(\nabla_\theta \mu_\theta^z(s_t, z_t)^\top \nabla_\theta \mu_\theta^z(s_t, z_t) \Upsilon^{-1}\right) \leq H_d^2 \cdot \left(\mathbf{1}^\top \mathrm{diag}\left(\Upsilon^{-1}\right)\right)
$$

$$
= H_d^2 \sum_{i=0}^{|a|-1} \frac{1}{\upsilon_i} = H_d^2 \, \mathrm{tr}(\Upsilon^{-1}). \tag{27}
$$

Using equation 27 the bound of the variance of the single gradient estimator is

$$
\mathrm{Var}\left[G(\bar\tau)\tilde{f}(\bar\tau)\right] \leq \frac{R^2(1-\gamma^T)^2}{(1-\gamma)^2}\left(\sum_{t=0}^{T-1} \mathbb{E}_{\bar\tau}\left[\mathrm{tr}\left(\nabla_\theta \mu_\theta^a(s_t, z_t)^\top \nabla_\theta \mu_\theta^a(s_t, z_t) \Sigma^{-1}\right)\right]\right.
$$

$$
\left. + \mathbb{E}_{\bar\tau}\left[\mathrm{tr}\left(\nabla_\theta \mu_\theta^z(s_t, z_t)^\top \nabla_\theta \mu_\theta^z(s_t, z_t) \Upsilon^{-1}\right)\right]\right)
$$

$$
\leq \frac{R^2(1-\gamma^T)^2}{(1-\gamma)^2} \sum_{t=0}^{T-1}\left(F_d^2 \, \mathrm{tr}(\Sigma^{-1}) + H_d^2 \, \mathrm{tr}(\Upsilon^{-1})\right)
$$

$$
= \frac{R^2(1-\gamma^T)^2 T}{(1-\gamma)^2}\left(F_d^2 \, \mathrm{tr}(\Sigma^{-1}) + H_d^2 \, \mathrm{tr}(\Upsilon^{-1})\right).
$$

Therefore, the gradient of the S2PG estimator in the diagonal Gaussian setting is

$$
\mathrm{Var}\left[\nabla_\theta \hat{J}_{\mathrm{S2PG}}(\Theta)\right] \leq \frac{R^2 \, \mathrm{tr}(\Sigma^{-1})(1-\gamma^T)^2 T}{N(1-\gamma)^2}\left(F_d^2 + H_d^2 \frac{\mathrm{tr}(\Upsilon^{-1})}{\mathrm{tr}(\Sigma^{-1})}\right)
$$

concluding the proof. $\qquad\square$

Notice that, when the action is a scalar, we get

$$
\mathrm{Var}\left[\nabla_\theta \hat{J}_{\mathrm{S2PG}}(\Theta)\right] \leq \frac{R^2(1-\gamma^T)^2 T}{N(1-\gamma)^2\sigma^2}\left(F_d^2 + H_d^2 \, \mathrm{tr}(\Upsilon^{-1})\sigma^2\right),
$$

matching closely both the previous results of Papini et al. (2022)— with the addition of the term depending on the internal state variance —and the special scalar action case of the generic bound derived with a full covariance matrix.

# B QUALITATIVE COMPARISON OF THE GRADIENT ESTIMATORS

In this section, we analyze the difference between the stochastic gradient estimator for stateful policies and the BPTT approach in greater detail.

In BPTT, the gradient needs to be propagated back to the initial state. This requires the algorithm to store the history of execution up to the current timestep, if we want to compute an unbiased estimate of the policy gradient. The common approach in the literature is to truncate the gradient propagation for a fixed history length. This approach is particularly well-suited if the recurrent policy does not require to remember long-term information.

Instead, our approach incorporates the learning of the policy state transitions into the value function, compensating for the increased variance of the policy gradient estimate (coming from the stochasticity of the policy state kernel) and allowing for the parallelization of the gradient computation. As shown in Figure 5, the stochastic gradient for stateful policies only uses local information at timestep $t$ to perform the update, while the BPTT uses the full history until the starting state $s_0$. The choice between the two estimators is non-trivial: a high dimensionality of the policy state may cause the Q-function estimation problem challenging. However, in practical scenarios, tasks can be solved with a relatively low-dimensional policy state vector. In contrast, a long trajectory implies multiple applications of the chain rule, that may consequently produce exploding or vanishing gradients, causing issues during the learning. This problem is not present when using stochastic stateful policies.

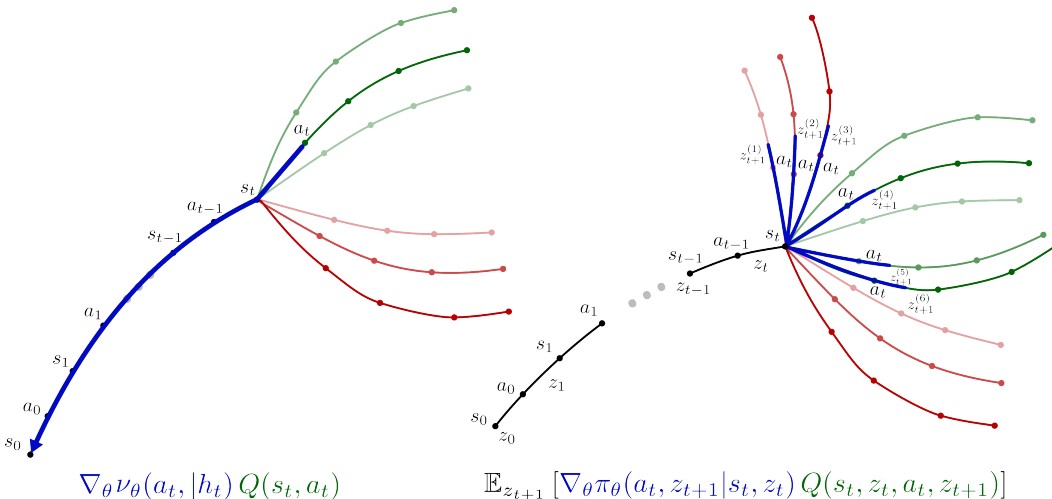

$$\nabla_\theta \nu_\theta(a_t, |h_t) \, Q(s_t, a_t) \qquad \mathbb{E}_{z_{t+1}} \left[ \nabla_\theta \pi_\theta(a_t, z_{t+1}|s_t, z_t) \, Q(s_t, z_t, a_t, z_{t+1}) \right]$$

Figure 5: Comparison of the gradient at the state $s_t$ and action $a_t$ in BPTT –left – and our stochastic gradient estimator – right– using the PGT. The red and green colors represents the value of future state, where green means high $Q$-value and red means low $Q$-value.

## C    ALGORITHM PSEUDOCODE

In the following, we present the pseudocode for SAC-RS, TD3-RS and PPO-RS algorithms. The algorithms are a straightforward modification of the vanilla SAC, TD3, and PPO approaches to support the internal policy state. In all cases, the algorithm receives in input a dataset of interaction (in our experiments one single step for SAC-RS and TD3-RS, while for PPO-RS is a batch of experience composed of multiple trajectories) coming from the current policy rollouts on the environment. In the pseudocode below we denote the parameter of the policy as $\boldsymbol{\theta}$, the target policy parameters as $\check{\boldsymbol{\theta}}$. Furthermore, we name the parameters of the i-th value function as $\psi_i$ and the corresponding target network $\check{\psi}_i$. In the following, we use the square bracket notation to denote the component of a vector. In this context, we refer to the vector of states, actions, and internal states inside the dataset (seen as an ordered list).

---

**Algorithm 1** TD3-RS

---

1: **function** UPDATETD3($\mathcal{D}$)
2:     $M$.ADD($\mathcal{D}$)                    ▷ Add the dataset of transitions $\mathcal{D}$ to the replay memory $M$
3:     **if** SIZE($M$) $> S_{\min}$ **then**              ▷ Wait until the replay memory has $S_{\min}$ transitions
4:         Sample a minibatch $\mathcal{B}$ from $M$
5:         $\hat{q} \leftarrow$ COMPUTETARGET($\mathcal{B}$)
6:         $Q_{\boldsymbol{\psi}_i}$.FIT($\mathcal{B}, \hat{q}$), for each $i \in \{0, 1\}$
7:         **if** it mod $D = 0$ **then**                      ▷ Perform the policy update every $D$ iterations
8:             Compute the loss

$$L_{\boldsymbol{\theta}}(\mathcal{B}) = -\frac{1}{N} \sum_{s \in \mathcal{B}} Q_{\boldsymbol{\psi}_0}(s, z, \mu_{\boldsymbol{\theta}}^a(s), \mu_{\boldsymbol{\theta}}^z(s))$$

9:             $\boldsymbol{\theta} \leftarrow$ OPTIMIZE($L_{\boldsymbol{\theta}}(\mathcal{B}), \boldsymbol{\theta}$)
10:         **end if**
11:         $\check{\theta} \leftarrow \tau\boldsymbol{\theta} + (1 - \tau)\check{\boldsymbol{\theta}}$
12:         $\check{\psi}_i \leftarrow \tau\boldsymbol{\psi}_i + (1 - \tau)\check{\psi}_i$
13:         it $\leftarrow$ it $+ 1$                                         ▷ Update the iteration counter
14:     **end if**
15:     **return** $\boldsymbol{\theta}$
16: **end function**

17: **function** COMPUTETARGET($\mathcal{B}$)
18:     **for** $(s, z, a, r, z', s') \in \mathcal{B}$ **do**
19:         **if** $s'$ is absorbing **then**
20:             $v_{\text{next}}(s', z') \leftarrow 0$
21:         **else**
22:             $a', z'' \leftarrow \mu_{\check{\boldsymbol{\theta}}}(s', z')$
23:             Sample $\epsilon^a, \epsilon^z \sim \mathcal{N}(0, \sigma_\epsilon)$
24:             $\epsilon_{\text{clp}}^a \leftarrow$ CLIP($\epsilon^a, -\delta_\epsilon, \delta\epsilon$)
25:             $\epsilon_{\text{clp}}^z \leftarrow$ CLIP($\epsilon^z, -\delta_\epsilon, \delta\epsilon$)
26:             $a_{\text{smt}} \leftarrow a' + \epsilon_{\text{clp}}^a$
27:             $z_{\text{smt}} \leftarrow z'' + \epsilon_{\text{clp}}^z$
28:             $a_{\text{clp}} \leftarrow$ CLIP($a_{\text{smt}}, a_{\min}, a_{\max}$)
29:             $z_{\text{clp}} \leftarrow$ CLIP($z_{\text{smt}}, z_{\min}, z_{\max}$)
30:             $v_{\text{next}}(s', z') \leftarrow \min_i Q_{\check{\psi}_i}(s', z', a_{\text{clp}}, z_{\text{clp}})$
31:         **end if**
32:         $\hat{q}(s, z, a, z') \leftarrow r + \gamma v_{\text{next}}(s', z')$
33:     **end for**
34:     **return** $\hat{q}$
35: **end function**

---

**Algorithm 2** SAC-RS

---

1: **function** UPDATESAC($\mathcal{D}$)
2: $\quad M$.ADD($\mathcal{D}$)
3: $\quad$ **if** SIZE($M$) > $S_{\min}$ **then**
4: $\quad\quad$ Sample a minibatch $\mathcal{B}$ from $M$
5: $\quad\quad \hat{q} \leftarrow$COMPUTESOFTTARGET($\mathcal{B}$)
6: $\quad\quad$ **if** SIZE($M$) > $S_{\text{warm}}$ **then** $\qquad\qquad$ ▷ Perform the policy update after $S_{\text{warm}}$ samples
7: $\quad\quad\quad$ Sample $a', z' \sim \pi_{\boldsymbol{\theta}}(\cdot|s,z) \; \forall (s,z) \in \mathcal{B}$
8: $\quad\quad\quad$ Compute the policy loss

$$L_{\boldsymbol{\theta}}(\mathcal{B}) = \alpha^a \log \pi_{\boldsymbol{\theta}}^a(a'|s,z) + \alpha^z \log \pi_{\boldsymbol{\theta}}^z(z'|s,z) - \frac{1}{N} \sum_{s \in \mathcal{B}} \min_i Q_{\boldsymbol{\psi}_i}(s,z,a',z')$$

9: $\quad\quad\quad \boldsymbol{\theta} \leftarrow$ OPTIMIZE($L_{\boldsymbol{\theta}}(\mathcal{B}), \boldsymbol{\theta}$)
10: $\quad\quad\quad$ Compute the $\alpha^a$ and $\alpha^z$ losses with target entropies $\bar{\mathcal{H}}^a$ and $\bar{\mathcal{H}}^z$

$$L_{\alpha^a}(\mathcal{B}) = -\frac{1}{N} \sum_{(s,z,a) \in \mathcal{B}} \alpha^a \left( \log \pi_{\boldsymbol{\theta}}^a(a|s,z) + \bar{\mathcal{H}}^a \right)$$

$$L_{\alpha^z}(\mathcal{B}) = -\frac{1}{N} \sum_{(s,z,z') \in \mathcal{B}} \alpha^z \left( \log \pi_{\boldsymbol{\theta}}^z(z'|s,z) + \bar{\mathcal{H}}^z \right)$$

11: $\quad\quad\quad \alpha^a \leftarrow$ OPTIMIZE($L_{\alpha^a}(\mathcal{B}), \alpha$)
12: $\quad\quad\quad \alpha^z \leftarrow$ OPTIMIZE($L_{\alpha^z}(\mathcal{B}), \alpha$)
13: $\quad\quad$ **end if**
14: $\quad\quad \hat{q} \leftarrow$COMPUTESOFTTARGET($\mathcal{B}$)
15: $\quad\quad Q_{\boldsymbol{\psi}}$.FIT($\mathcal{B}, \hat{q}$)
16: $\quad\quad \check{\boldsymbol{\psi}}_i \leftarrow \tau \boldsymbol{\psi}_i + (1-\tau)\check{\boldsymbol{\psi}}_i$
17: $\quad$ **end if**
18: $\quad$ **return** $\boldsymbol{\theta}$
19: **end function**

20: **function** COMPUTESOFTTARGET($\mathcal{B}$)
21: $\quad$ **for** $(s,z,r,a',z',s') \in \mathcal{B}$ **do**
22: $\quad\quad$ **if** $s'$ is absorbing **then**
23: $\quad\quad\quad v_{\text{next}}(s',z') \leftarrow 0$
24: $\quad\quad$ **else**
25: $\quad\quad\quad$ Sample $a', z'' \sim \pi_{\boldsymbol{\theta}}(\cdot|s')$
26: $\quad\quad\quad h_{\text{bonus}} \leftarrow -\alpha^a \log \pi_{\boldsymbol{\theta}}(a'|s',z') - \alpha^z \log \pi_{\boldsymbol{\theta}}(z''|s',z')$
27: $\quad\quad\quad v_{\text{next}}(s',z') \leftarrow \min_i Q_{\check{\psi}_i}(s',z',a',z'') + h_{\text{bonus}}$
28: $\quad\quad$ **end if**
29: $\quad\quad \hat{q}(s,z,a,z') \leftarrow r + \gamma v_{\text{next}}(s',z')$
30: $\quad$ **end for**
31: $\quad$ **return** $\hat{q}$
32: **end function**

---

---

**Algorithm 3** PPO-RS

---

1: **function** UPDATEPPO($\boldsymbol{\theta}, \mathcal{D}$)
2:     $V', A \leftarrow$ COMPUTEGAE($V_{\boldsymbol{\psi}}, \mathcal{D}$)
3:     $V_{\boldsymbol{\psi}}$.FIT($\mathcal{D}, V'$)                                                          ▷ Update the value function
4:     **for** $i \leftarrow 0$ to $N$ **do**
5:         Split the dataset $\mathcal{D}$ in $K$ minibatches $\{\mathcal{B}_k | k \in [0, K-1]\}$
6:         **for** $k \leftarrow 0$ to $K$ **do**
7:             compute the surrogate loss on the minibatch

$$L_{\boldsymbol{\theta}}(\mathcal{B}_k) = \frac{1}{N} \sum_{(s,z,a,z') \in \mathcal{B}_k} \min \left( \frac{\pi_{\boldsymbol{\theta}}(a, z'|s, z)}{q(a, z'|s, z)} A(s, z, a, z'), \right.$$

$$\left. \text{clip} \left( \frac{\pi_{\boldsymbol{\theta}}(a, z'|s, z)}{q(a, z'|s, z)}, 1 - \varepsilon, 1 + \varepsilon \right) A(s, z, a, z') \right)$$

8:             $\boldsymbol{\theta} \leftarrow$ OPTIMIZE($L_{\boldsymbol{\theta}}(\mathcal{B}_k), \boldsymbol{\theta}$)
9:         **end for**
10:     **end for**
11:     **return** $\boldsymbol{\theta}$
12: **end function**

13: **function** COMPUTEGAE($V_{\boldsymbol{\psi}}, \mathcal{D}$)
14:     **for** $k \leftarrow 0 \dots \text{len}(\mathcal{D})$ **do**
15:         $v[k] \leftarrow V_{\psi}(s[k], z[k])$
16:         $v_{\text{next}}[k] \leftarrow V_{\psi}(s'[k], z'[k])$
17:     **end for**
18:     **for** $k_{\text{rev}} \leftarrow 0 \dots \text{len}(\mathcal{D})$ **do**
19:         $k \leftarrow \text{len}(\mathcal{D}) - k_{\text{rev}} - 1$
20:         **if** $s'[k]$ **is** last **then**
21:             **if** $s'[k]$ **is** absorbing **then**
22:                 $A(s[k], z[k], a[k], z'[k]) \leftarrow r[k] - v[k]$
23:             **else**
24:                 $A(s[k], z[k], a[k], z'[k]) \leftarrow r[k] + \gamma v_{\text{next}}[k] - v[k]$
25:             **end if**
26:         **else**
27:             $\delta \leftarrow A(s[k+1], z[k+1], a[k+1], z'[k+1])$
28:             $A(s[k], z[k], a[k], z'[k]) \leftarrow r[k] + \gamma v_{\text{next}}[k] - v[k] + \gamma \lambda \delta$
29:         **end if**
30:     **end for**
31:     **return** $v, A$
32: **end function**

---

# D    NETWORK STRUCTURES

In this section, we describe the structure of the networks used in this paper. Figure 6 presents all networks used for the policies and the critics in the RL setting. For the BPTT, the policy networks were used, yet the red paths were dropped. Also, the critics did not use the hidden states as inputs. For the IL setting, the networks from SAC are used for LS-IQ, and the networks from PPO are used for GAIL. For the recurrent networks, we generally use GRUs even though any other recurrent network, such as LSTMs, can also be used.

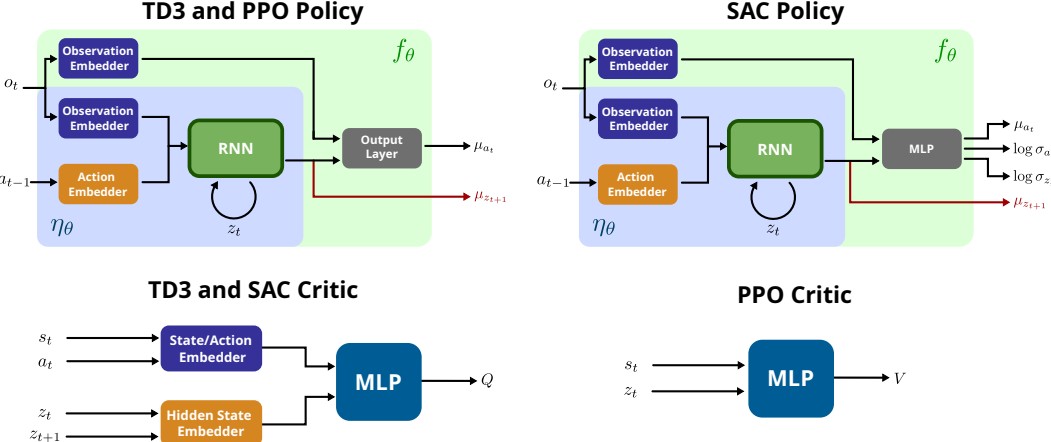

Figure 6: Network architectures used for the policies and critics in PPO-RS, TD3-RS, SAC-RS, PPO-BPTT, TD3-BPTT, and SAC-BPTT. For the BPTT variants, the red paths do not exist. For the S2PG approach, the internal state is sampled from the noisy distribution. These architectures of the policies were originally used in Ni et al. (2022).

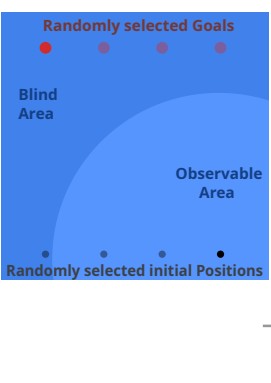 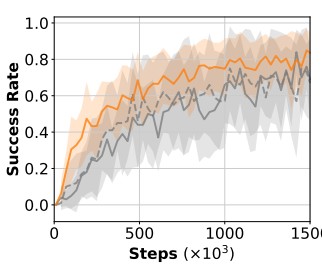

Figure 7: Point mass experiments. In the blind area, the goal is not visible. The task involves remembering its position. Initial states, goals, and blind areas are randomly sampled. SAC-RS is a recurrent-stochastic version of SAC, using S2PG.

——— SAC-BPTT-5    - - - SAC-BPTT-10    ——— SAC-RS

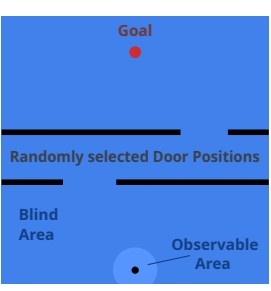 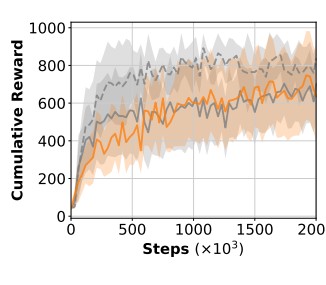

Figure 8: Point mass door experiments. In the blind area, the positions of the two doors are not visible. The task involves remembering its position, which are randomly sampled. SAC-RS is a recurrent-stochastic version of SAC, using S2PG.

——— SAC-BPTT-5    - - - SAC-BPTT-10    ——— SAC-RS

# E    ADDITIONAL EXPERIMENTS

## E.1    MEMORY TASK

These tasks are about moving a point mass in a 2D environment. They are shown in Figure 7 and 8. For the first task, a random initial state and a goal are sampled at the beginning of each episode. The agent reads the desired goal information while it is close to the starting position. Once the agent is sufficiently far from the starting position, the information about the goal position is zeroed-out. The task consists of learning a policy that remembers the target goal even after leaving the starting area. As the distance to the goal involves multiple steps, long-term memory capabilities are required. Similarly, the second task hides the position of two randomly sampled doors in a maze once the agent leaves the observable area. Once the agents touches the black wall, an absorbing state is reached, and the environment is reset. We test our approach in the privileged information setting using SAC. We compare our approach with SAC-BPTT with a truncation length of 5 and 10. For the first task, our approach – SAC-RS – outperforms BPTT, while it performs slightly weaker compared to BPTT with a truncation length of 10 on the second task. We expect that the reason for the drop in performance is caused by the additional variance of our method in combination with the additional absorbing states in the environment. The additional variance in our policy leads to increased exploration, which increases the chances of touching the wall and reaching an absorbing state. Nonetheless, our approach has a significantly shorter training time, analogously to the results shown in Figure 3, for both tasks.

## E.2    STATEFUL POLICIES AS INDUCTIVE BIASES

Adding inductive biases into the policy is a common way to impose prior knowledge into the policy Ijspeert et al. (2007); Bellegarda & Ijspeert (2022a); Al-Hafez & Steil (2021); Liu et al. (2021). To show that stateful policies are of general interest in RL – not only for POMDPs – we show in this section that they can be used to elegantly impose an arbitrary inductive bias directly into the policy. In contrast to commonly used biases, our approach allows learning the parameters of the inductive bias as well. We conduct experiments with a policy that includes oscillators –– coupled Central Pattern Generators (CPG) – to impose an oscillation into the policy. The latter is common

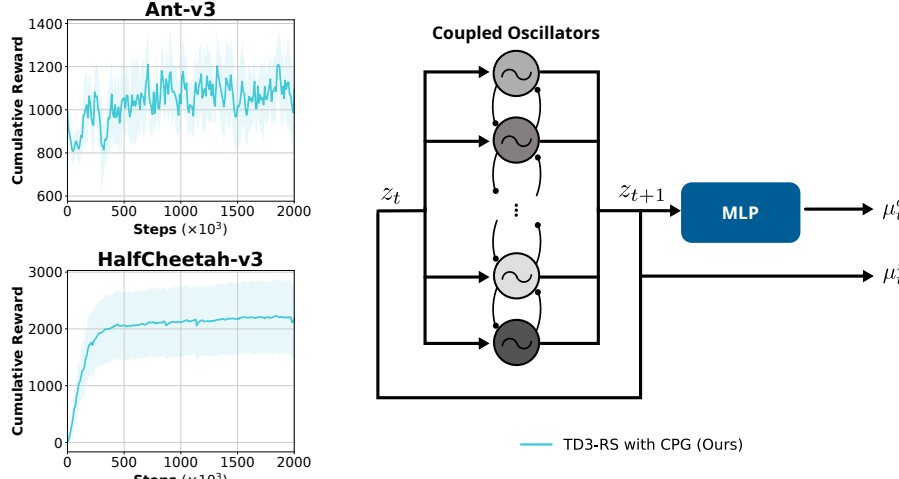

Figure 9: Experiments in which a set of coupled central pattern generators are used as a policy. The left side shows the cumulative reward on HalfCheetah and Ant. The right side shows the structure of the policy. Note that the policy does not use the environment state making it blind.

for locomotion tasks Miki et al. (2022). We model the oscillators as a set of ordinary differential equations and simulate the latter using the explicit Euler integrator. Figure 9 presents the results. Using our algorithm, we are able to train the parameters of the CPGs and an the additional network to learn simple locomotion skills on HalfCheetah and Ant. These policies solely rely on their internal state and do not use the environment state making them blind. With this simple experiment, we want to show that we can impose arbitrary dynamics on the policy and train the latter using our approach. Note that plain oscillators do not constitute a very good inductive bias. More sophisticated dynamical models that also take the environment state into account are needed to achieve better results.

**Related Work.** Shi et al. (2022) train CPGs using a black-box optimizer instead of BPTT, while simultaneously learning an RL policy for residual control. Cho1 et al. (2019) optimize a CPG policy using RL without BPTT by exposing the internal state to the environment but do not provide a theoretical derivation of the resulting policy gradient. Campanaro et al. (2021) train a CPG policy using PPO without BPTT. However, the authors neglect the influence of the internal state on the value function.

### E.3 COMPARISON ON STANDARD POMDP BENCHMARKS

In the following, we evaluate the two gradient estimators in standard MuJoCo POMDP benchmarks, in the on-policy setting. These tasks are taken from Ni et al. (2022) and hide the velocity. All experiments use the PPO algorithm as a learning method. Figure 10 presents the results. Our approach outperforms BPTT in all benchmarks in terms of the number of used samples and time. Unfortunately, using on-policy approaches on these benchmarks, we do not achieve satisfactory performances in all environments (e.g. Hopper and Walker). This is probably due to the increased difficulty of exploring with partial observation, rather than an issue in gradient estimation. Indeed, in the other tasks, where more information is available, or the exploration is less problematic – e.g., the failure state is more difficult to reach – we outperform the baseline. It should be noted that the performance gain in terms of computation time is not massive: this is due to the reduced number of gradient computations in the on-policy scenario, which is performed in batch after many environment steps. Table 1 presents the training time needed to run 1 million environment steps for all versions of PPO. Accordingly, table 2 illustrates the training time factors for each PPO version in comparison to the vanilla variant.

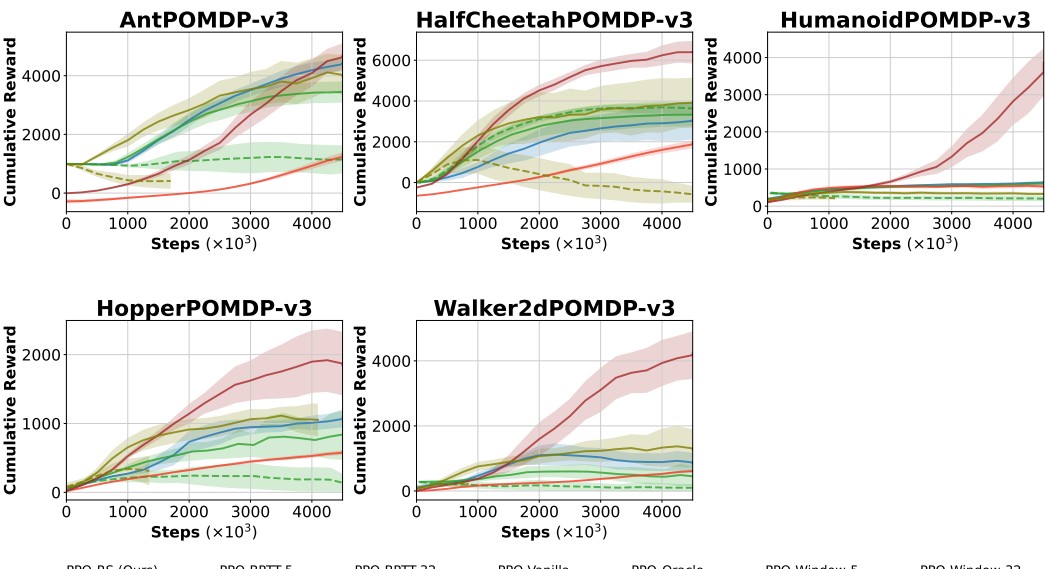

Figure 10: Comparison of PPO with the proposed stochastic policy state kernel and PPO with BPTT on fully partially observable MuJoCo Gym Tasks. The number behind the BPTT implementations indicates the truncation length. Abscissa shows the cumulative reward. Ordinate shows the number of training steps ($\times 10^3$).

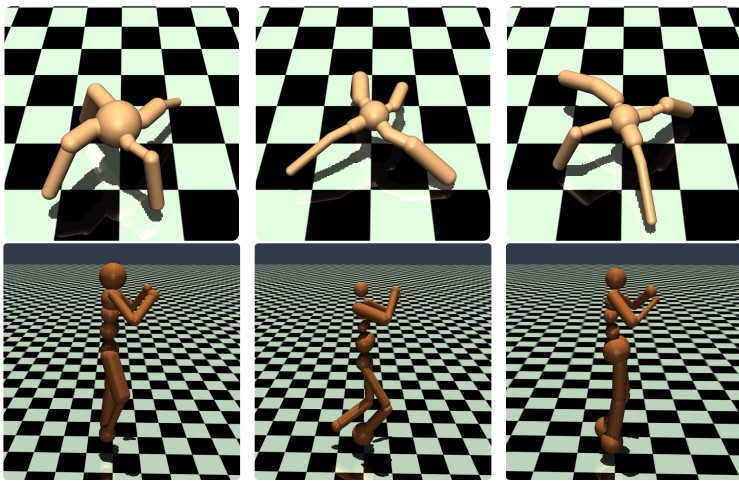

Figure 11: Example images of the Mujoco Gym randomized mass environment used within this work. The upper row shows three randomly sampled Ant environments, while the lower row show three Humanoid environments.

### E.4 COMPARISON ON ROBUSTNESS BENCHMARKS WITH PRIVILEGE INFORMATION

In this section, we evaluate the performance of S2PG against BPTT on the POMDP setting with additional randomized masses. For the latter, masses of each link in a model are randomly sampled in a ±30% range. Figure 11 shows examples of randomly sampled Ant and Humanoid environments. Here, we consider the scenario where the critic has privileged information (knowledge of velocities and mass distributions), while the policy uses only partial observability.

Figure 12 presents the reward plots. Figure 3 also presents the average runtime of each approach used for these tasks. The experiment run for a maximum of two million steps and four days of computation. The results do not show an approach clearly outperforming the others in all tasks. S2PG based approaches seem to work more robustly in high-dimensional tasks, such as AntPOMDP-v3 and HumanoidPOMDP-v3. Curiously, the BPTT version of TD3 is not able to learn in the AntPOMDP-v3 task. In general, S2PG seems to struggle in the HalfCheetahPOMDP-v3, HopperPOMDP-v3 and in WalkerPOMDP-v3 tasks. To better compare the performance of all approaches on the low-dimensional tasks – Hopper, Walker and HalfCheetah – and the high-dimensional tasks – Humanoid and Ant –, we presents the averaged performance plots in Figure 13. As can be seen, our approach performs significantly better on the high-dimensional tasks while being worse on the low-dimensional ones. We believe that the reason for the worse performance of BPTT on high-dimensional tasks could be the potential risk of exploding gradients, which in turn cause exploding variance (c.f., Section 3). In contrast, we believe that the worse performance of our method on low-dimensional tasks can be traced back to the increased amount of variance in the policy in conjunction with the presence of absorbing states in the case of Hopper and Walker, similar to the memory tasks. Table 1 presents the training time needed to run 1 million environment steps for all versions of SAC and TD3. Accordingly, table 2 illustrates the training time factors for each version in comparison to the vanilla variant.

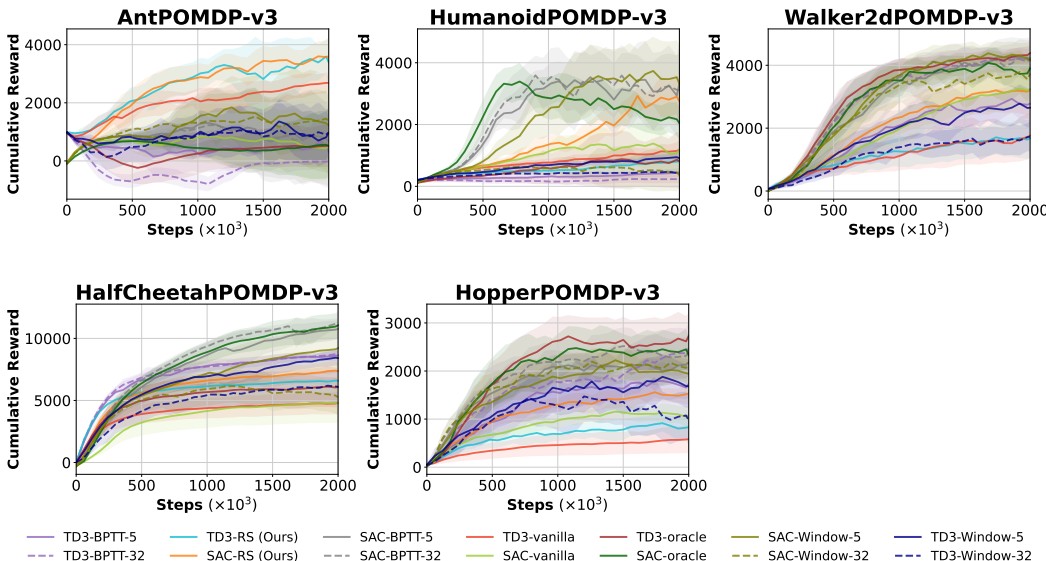

Figure 12: SAC and TD3 results on the MuJoCo Gym Tasks with partially observable policies and privileged critics. Additionally, masses of each link are randomly sampled at the beginning of each episode. The number behind the BPTT implementations indicates the truncation length. Abscissa shows the cumulative reward. Ordinate shows the training time in hours.

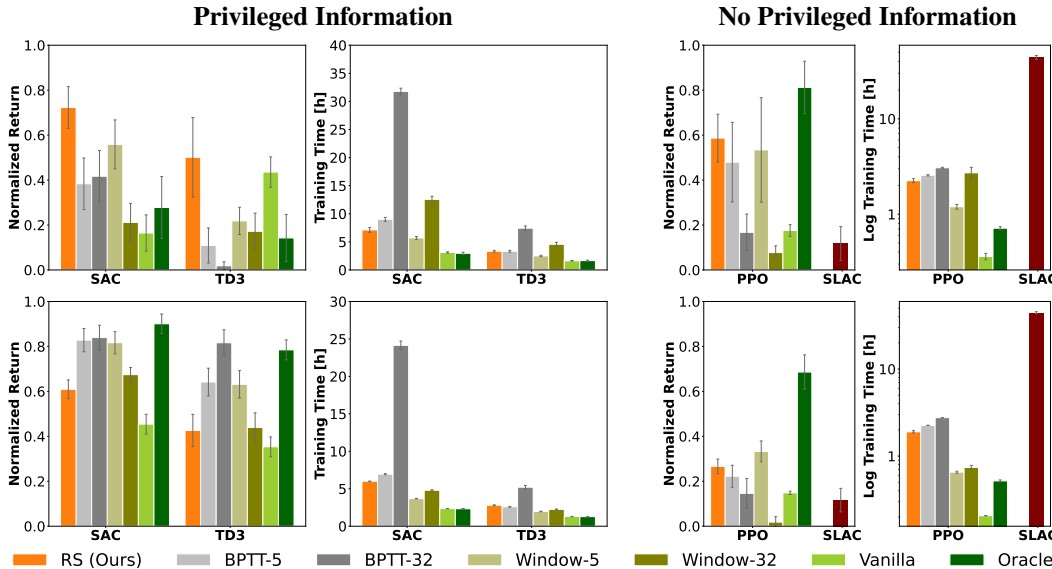

Figure 13: Results show the mean and 95% confidence interval of the normalized return and the training time needed for 1 million steps across high-dimensional – **TOP** row; Ant and Humanoid – and low-dimensional – **BOTTOM** row; Hopper, Walker and HalfCheetah – POMDP Gym tasks. Comparison of different RL agents on POMDP tasks using our stateful gradient estimator, BPTT with a truncation length of 5 and 32, the SLAC algorithm and stateless versions of the algorithms using a window of observation with length 5 and 32. The "Oracle" approach is a vanilla version of the algorithm using full-state information.

Table 1: Runtime comparison of different algorithms on all reinforcement learning tasks. For SAC and TD3, privileged information for the critic is used, and the tasks include randomization of the masses and occlusion of the velocities, while for PPO no privileged information is used and only occlusion of the velocities is used for the tasks. The window, vanilla, and oracle approaches are grayed out as they do not constitute stateful policies. The bold values indicate the lowest training time across all stateful algorithms. All values are computed by taking the mean over 10 seeds and computing the 95% confidence interval.

| | | **Time in hours needed for 1 million environment steps [lower is better]** | | | | | | |
| | | **BPTT** | | **Window** | | **Vanilla** | **Oracle** | **RS (Ours)** |
| | **Env.** | **5** | **32** | **5** | **32** | | | |
| | **Stateful Policy** | yes | yes | no | no | no | no | yes |
| SAC | Ant | $7.8 \pm 0.067$ | $30.4 \pm 1.018$ | $4.8 \pm 0.120$ | $10.6 \pm 0.548$ | $2.8 \pm 0.047$ | $2.5 \pm 0.092$ | **6.2 ± 0.256** |
| | Hopper | $6.8 \pm 0.168$ | $22.3 \pm 0.406$ | $3.5 \pm 0.036$ | $4.5 \pm 0.089$ | $2.3 \pm 0.026$ | $2.2 \pm 0.070$ | **5.9 ± 0.017** |
| | HalfCheetah | $6.9 \pm 0.291$ | $27.1 \pm 0.361$ | $3.7 \pm 0.054$ | $5.1 \pm 0.106$ | $2.3 \pm 0.071$ | $2.4 \pm 0.010$ | **6.0 ± 0.041** |
| | Humanoid | $10.1 \pm 0.285$ | $32.9 \pm 1.317$ | $6.5 \pm 0.118$ | $14.1 \pm 0.478$ | $3.4 \pm 0.049$ | $3.3 \pm 0.079$ | **7.9 ± 0.027** |
| | Walker2d | $6.8 \pm 0.202$ | $22.7 \pm 0.696$ | $3.7 \pm 0.051$ | $4.6 \pm 0.104$ | $2.4 \pm 0.034$ | $2.3 \pm 0.067$ | **6.0 ± 0.026** |
| TD3 | Ant | **3.0 ± 0.019** | $6.6 \pm 0.119$ | $2.3 \pm 0.017$ | $3.8 \pm 0.127$ | $1.5 \pm 0.018$ | $1.5 \pm 0.010$ | $3.0 \pm 0.124$ |
| | Hopper | **2.6 ± 0.058** | $4.5 \pm 0.176$ | $1.9 \pm 0.054$ | $2.1 \pm 0.030$ | $1.3 \pm 0.025$ | $1.3 \pm 0.018$ | $2.8 \pm 0.093$ |
| | HalfCheetah | **2.6 ± 0.010** | $5.8 \pm 0.089$ | $2.0 \pm 0.086$ | $2.4 \pm 0.070$ | $1.3 \pm 0.022$ | $1.2 \pm 0.042$ | $2.7 \pm 0.100$ |
| | Humanoid | $3.6 \pm 0.042$ | $8.1 \pm 0.325$ | $2.7 \pm 0.047$ | $5.2 \pm 0.165$ | $1.8 \pm 0.030$ | $1.8 \pm 0.022$ | **3.6 ± 0.053** |
| | Walker2d | **2.4 ± 0.006** | $5.1 \pm 0.242$ | $1.9 \pm 0.018$ | $2.2 \pm 0.062$ | $1.3 \pm 0.024$ | $1.3 \pm 0.012$ | $2.9 \pm 0.052$ |
| PPO | Ant | $2.8 \pm 0.001$ | $3.3 \pm 0.001$ | $0.7 \pm 0.088$ | $1.6 \pm 0.079$ | $0.4 \pm 0.005$ | $0.6 \pm 0.007$ | **2.1 ± 0.036** |
| | Hopper | $2.5 \pm 0.001$ | $3.1 \pm 0.001$ | $0.5 \pm 0.004$ | $0.5 \pm 0.006$ | $0.3 \pm 0.006$ | $0.4 \pm 0.004$ | **1.8 ± 0.061** |
| | HalfCheetah | $2.5 \pm 0.001$ | $3.1 \pm 0.001$ | $0.5 \pm 0.003$ | $0.6 \pm 0.003$ | $0.4 \pm 0.012$ | $0.4 \pm 0.001$ | **2.2 ± 0.044** |
| | Humanoid | $2.9 \pm 0.001$ | $3.5 \pm 0.001$ | $3.5 \pm 0.001$ | $3.5 \pm 0.001$ | $0.5 \pm 0.052$ | $0.7 \pm 0.009$ | **2.1 ± 0.033** |
| | Walker2d | $2.5 \pm 0.001$ | $3.1 \pm 0.001$ | $0.5 \pm 0.004$ | $0.5 \pm 0.008$ | $0.3 \pm 0.022$ | $0.4 \pm 0.003$ | **1.8 ± 0.041** |

Table 2: Runtime comparison of different algorithms on all reinforcement learning tasks. For SAC and TD3, privileged information for the critic is used, and the tasks include randomization of the masses and occlusion of the velocities, while for PPO no privileged information is used and only occlusion of the velocities is used for the tasks. The window, vanilla, and oracle approaches are grayed out as they do not constitute stateful policies. The factors are computed for 1 million environment steps. The bold values indicate the lowest factor across all stateful algorithms. All values are computed by taking the mean over 10 seeds.

| | | **Factor of time increase compared to Vanilla version [lower is better]** | | | | | |
| | | **BPTT** | | **Window** | | **Vanilla** | **RS (Ours)** |
| | **Env.** | **5** | **32** | **5** | **32** | | |
| | **Stateful Policy** | yes | yes | no | no | no | yes |
| SAC | Ant | 2.804 | 10.935 | 1.730 | 3.822 | 1.000 | **2.241** |
| | Hopper | 2.935 | 9.599 | 1.497 | 1.956 | 1.000 | **2.531** |
| | HalfCheetah | 3.022 | 11.859 | 1.614 | 2.213 | 1.000 | **2.610** |
| | Humanoid | 2.936 | 9.580 | 1.887 | 4.106 | 1.000 | **2.290** |
| | Walker2d | 2.878 | 9.597 | 1.578 | 1.948 | 1.000 | **2.526** |
| TD3 | Ant | **2.024** | 4.535 | 1.546 | 2.620 | 1.000 | 2.056 |
| | Hopper | **2.012** | 3.476 | 1.489 | 1.643 | 1.000 | 2.146 |
| | HalfCheetah | **2.069** | 4.606 | 1.555 | 1.864 | 1.000 | 2.120 |
| | Humanoid | 2.037 | 4.555 | 1.512 | 2.896 | 1.000 | **2.024** |
| | Walker2d | **1.913** | 4.024 | 1.527 | 1.724 | 1.000 | 2.245 |
| PPO | Ant | 7.845 | 9.457 | 2.120 | 4.637 | 1.000 | **5.815** |
| | Hopper | 9.128 | 11.194 | 1.706 | 1.825 | 1.000 | **6.602** |
| | HalfCheetah | 6.888 | 8.454 | 1.331 | 1.536 | 1.000 | **6.023** |
| | Humanoid | 6.018 | 7.183 | 7.183 | 7.183 | 1.000 | **4.264** |
| | Walker2d | 8.735 | 10.698 | 1.666 | 1.829 | 1.000 | **6.048** |

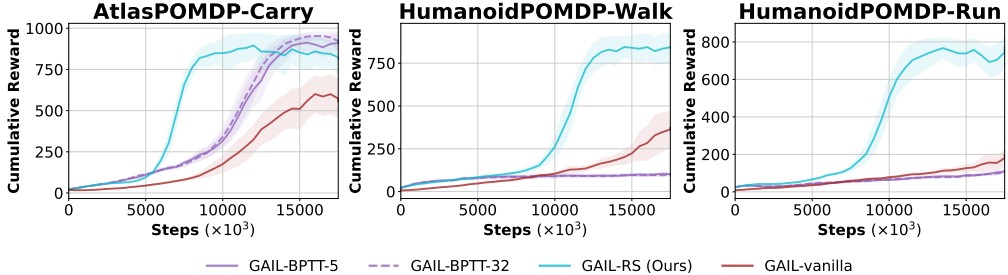

Figure 14: GAIL results on all IL tasks with partially observable policies and privileged critics. The number behind the BPTT implementations indicates the truncation length. Abscissa shows the cumulative reward. Ordinate shows the number of training steps ($\times 10^3$).

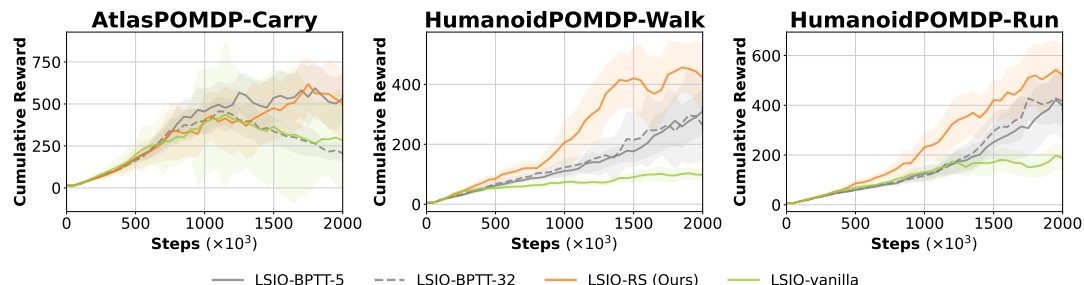

Figure 15: LS-IQ results on all IL tasks with partially observable policies and privileged critics. The number behind the BPTT implementations indicates the truncation length. Abscissa shows the cumulative reward. Ordinate shows the number of training steps ($\times 10^3$).

## E.5 IMITATION LEARNING EXPERIMENTS

In this section, we present further results of the IL tasks from the main paper. As can be seen in Figure 14 and 15, our results show that the S2PG version of GAIL and LS-IQ – GAIL-RS and LSIQ-RS – are competitive with the BPTT variants, even when the truncation length is increased. In the HumanoidPOMDP-Walk task, GAIL-RS is even able to outperform by a large margin the BPTT variant. As stressed in the paper, the advantage of using S2PG over BPTT is the reduced computation time. As LS-IQ runs SAC in the inner RL loop, and GAIL runs PPO, the runtimes of these algorithms are very similar to their RL counterparts. For the Atlas experiments, it can be seen that the vanilla policy in GAIL is already performing well, despite being outperformed by our method. This shows that this task does not require much memory. In contrast, the humanoid tasks require much more memory, as can be seen by the poor results of the vanilla approaches. Interestingly, we found the GAIL-BPTT variants are often very difficult to train on these tasks, which is the reason why they often result in unstable learning.

