# OpenReview forum: "Time-Efficient Reinforcement Learning with Stochastic Stateful Policies"
_ICLR.cc/2024/Conference — ICLR 2024 poster_

### Official Review · Reviewer_wQTz · 2023-10-30

**Soundness:** 3 good
**Presentation:** 1 poor
**Contribution:** 3 good
**Rating:** 6
**Confidence:** 4

**Summary:**

The paper presents a new method entitled 'stochastic stateful policies' for performing reinforcement learning (RL) with policies having an internal state. The method can be applied to POMDP settings and other non-Markovian setups where it is necessary to have access to the whole trajectory, not only the current state. The method is compared against the state-of-the-art backpropagation through time (BPTT) approach and is more computationally efficient but may result in higher variance. Fundamental theorems are provided, and the method is evaluated in a series of problems, showing improvement, especially in higher dimensional cases.

**Strengths:**

Designing new RL approaches tailored for the POMDP setting is important, as a practical deployment will certainly be non-Markovian. To my knowledge, the existing state-of-the-art is to use existing RL algorithm with stateful actors/critics, e.g., along the current state, a history encoded using an LSTM/RNN/GRU is provided.

The paper has much content; the theoretical section is rich, with a few fundamental results -- the policy gradient theorem, the deterministic policy gradient theorem, and IMHO, the most important - a variance upper bounds are provided in the studied setting. However, it mainly adapts existing results to the current setting, which is unsurprising. Clean proofs of the theorems are provided.
The experimental part is also rich; a few experiments reported (unfortunately mostly in the appendix) demonstrating the approach's potential. I think it is fair to say that the paper contains a thorough experimental study.

The method performs better with less computational and memory burden than the SOTA BPTT approach in high-dimensional problems. Moreover, the authors are honest, and some settings in which the method does not perform well (likely due to increased variance) are also provided, which is nice.

Overall, after reading the main part of the paper, I saw the paper as a borderline paper. However, luckily, after going through the much longer appendix, I became more positive about the paper. Let me note here that as a reviewer, I am not obliged to make a detailed pass over the paper appendix and should base my judgment upon the main part content.

Considering the whole content, this is a convincing paper. However, there is a danger if a reader goes through the main part only may not appreciate the results fully. This brings my main concern about the paper - I am not sure if the authors took significant effort in making the main part of the paper stand-alone and convincing enough, considering the amount and quality of produced results. The main part needs a major revision before publication. I will reconsider my score after my concerns are addressed in the rebuttal phase. See detailed remarks and questions below.

**Weaknesses:**

My main concern with the paper is that the main part needs to present the available results self-contained and convincing enough, which is not the case. Before accepting the paper, it will need thorough edits addressing the concerns and the questions I present below.

1. The Entire Theoretical part, especially the equations, should be formatted more concisely; there is no reason to display the equations in two lines. Notably, only only circa. 1.5 pages in the main part are left for the experimental results. Only two experiments are presented, and there are a few interesting ones in the appendix. The 'Stateful policies as inductive biases' experiment is especially interesting, and it is also mentioned in the introduction.

2. The main advantage of the approach over BPTT - improved computational efficiency is not shown in the paper. It would strengthen the paper when the computational benefits are demonstrated, using at least a table with actual wall-times comparison. It is only mentioned in the paper without proof: "The overall results show that our approach has major computational benefits w.r.t BPTT with long histories at the price of a slight drop in asymptotic performance for SAC and TD3".

3. Often, results are mentioned in the main part with a reference to figures in the appendix. I emphasize that most readers will likely stop reading after the main part, so it would be nice to have at least a 'teaser' of the results in the main part.

**Questions:**

### Main questions
* Provide a wall-time comparison of the introduced S2PG approach with (as I understand more time-consuming) BPTT approach.
* Used assumptions in the theoretical part - argue their physical motivation and preferably cite some established works that introduced them,
* Hiding velocity experiment - this will not check the benefits of any long-term history encoding, as only two states can be used to approximate velocity.
* I have a suggestion for another experiment - simplify further the policy by removing the $\pi_\theta^a$ , and keeping only $\pi_\theta^z$.
* What do you mean by  "our method could be used in combination with complex policy structures such as Neural ODEs." can you elaborate?

### Minor remarks/questions
* p. 3 eq. (1) provide formula for $J(\tau)$;
* p. 4, caption Fig. 1 Do you rather mean 'from left to right' ?
* p. 4 'equations equation' -> 'equation;
* p. 6 "causality dilemma: On one hand, the policy can learn a condensed representation of the
history assuming accurate Q-value estimates. On the other hand, the Q-function can be learned using bootstrapping techniques, assuming a reliable condensed representation of the history. But doing
both at the same time often results in unstable training." be more clear here;
* p. 6 "We extend the letter" -> the latter;
* p. 8 caption Fig. 2 "for 1 Mio. steps" -> what is Mio. ?

---

> ### Author Response · Authors · 2023-11-15
> **Response to Reviewer wQTz**
>
> We very much appreciate the valuable suggestions of the reviewer and are especially grateful for the time invested by the reviewer in reading also the appendix. We agree with the reviewer that the old experiment section was not self-contained. In the updated version of the paper, we updated the experiment section to include 'at least a teaser' of all experiments; an idea from the reviewer we very much liked. Now. the reader gets an overview of all experiments without the need to dive into the appendix.
>
> We have highlighted all changes in the paper in blue to make it easier for the reviewer to spot the differences. In the following, we answer to the specific concerns of the reviewer.
>
> >The Entire Theoretical part, especially the equations, should be formatted more concisely; there is no reason to display the equations in two lines. Notably, only only circa. 1.5 pages in the main part are left for the experimental results. Only two experiments are presented, and there are a few interesting ones in the appendix. The 'Stateful policies as inductive biases' experiment is especially interesting, and it is also mentioned in the introduction.
>
> We updated the paper and shortened all two-line equations to single-line equations in the main part. We also shortened the theoretical part to make more space for experiments. We included a new figure, which gives an overview of all experiments. Every experiment is now discussed in a separate paragraph, including the inductive bias experiment. In doing so, all the highlights are in the main paper. If the reviewer thinks that there is still an important experiment missing in the main paper from the appendix, we would be happy to add it.
>
> >The main advantage of the approach over BPTT - improved computational efficiency is not shown in the paper. It would strengthen the paper when the computational benefits are demonstrated, using at least a table with actual wall-times comparison. It is only mentioned in the paper without proof: "The overall results show that our approach has major computational benefits w.r.t BPTT with long histories at the price of a slight drop in asymptotic performance for SAC and TD3".
>
> The computation times were shown in Figure 2 (now Figure 3) together with the rewards. The diagram is labeled with 'Training Time [h]'. We agree that this was not clear from the main text in the old version. We now specifically refer to it in the main text. The training time plots show the time needed for 1 million steps *averaged across all POMDP Gym tasks*. We now also added detailed tables with specific training times for each environment in the Appendix. Note that the differences in training times in the imitation learning setting are comparable to the reinforcement learning results, as LS-IQ is based on SAC and Gail is based on PPO.
>
> >Often, results are mentioned in the main part with a reference to figures in the appendix. I emphasize that most readers will likely stop reading after the main part, so it would be nice to have at least a 'teaser' of the results in the main part.
>
> All experiments from our paper are now presented in the main part. We absolutely agree with the reviewer that the experiments in the main sections were not self-contained before. We now give an overview of all tasks with a new figure and explain every experiment in a separate paragraph. In doing so, we reduced the amount of references to the appendix and made the main paper more self-contained.
>
> >Provide a wall-time comparison of the introduced S2PG approach with (as I understand more time-consuming) BPTT approach.
>
> As mentioned before, this is now more clear in the paper.
>
> >Used assumptions in the theoretical part - argue their physical motivation and preferably cite some established works that introduced them
>
> The assumptions used in the variance analysis are taken from prior work [1] and extended to the matrix setting. They basically assume a well-behaved policy structure by bounding the maximum magnitude of the gradients. The assumptions on the MPD used in the proofs of the policy gradient theorems are directly taken from prior work [2]. Similarly, they assume that the MPD is well-behaved, resulting in an MDP without singularities and/or explosion. We added the respective citations in the paper.
>
> [1] Tingting Zhao, Hirotaka Hachiya, Gang Niu, and Masashi Sugiyama. Analysis and improvement of
> policy gradient estimation. In Proceeding of the Twenty-fifth Conference on Neural Information
> Processing Systems, Granada, Spain, December 2011.
>
> [2] David Silver, Guy Lever, Nicolas Heess, Thomas Degris, Daan Wierstra, and Martin Riedmiller. De-
> terministic policy gradient algorithms. In Proceeding of the International Conference on Machine
> Learning, Beijing, China, June 2014.

---

> > ### Author Response · Authors · 2023-11-15
> > **Further Response to Reviewer wQTz**
> >
> > >Hiding velocity experiment - this will not check the benefits of any long-term history encoding, as only two states can be used to approximate velocity.
> >
> > We totally agree just that hiding the velocities is not a reasonable task to check for long-term histories. This task was taken from prior work [3].  However, note that for the privileged information setting -- i.e., all TD3 and SAC experiments -- we also randomized the masses of all links of a body by $\pm 30\\%$, resulting in a very challenging task.
> >
> > [3] Tianwei Ni, Benjamin Eysenbach, and Ruslan Salakhutdinov. Recurrent model-free rl can be a
> > strong baseline for many pomdps. In ICML 2022, pp. 16691–16723.
> >
> > >I have a suggestion for another experiment - simplify further the policy by removing the $\pi^a_\theta$, and keeping only  $\pi^z_\theta$
> >
> > We are not sure if we fully understand this idea. Do you mean that we should define the next policy state to also be the action? This is something that is totally doable but limits the policy state to representations useful for actions.
> >
> > >What do you mean by "our method could be used in combination with complex policy structures such as Neural ODEs." can you elaborate?
> >
> > In our main paper, we now discuss and highlight how a parametric ODE can be placed in a policy and directly optimized by our gradient estimator (c.f., Fig 2). In our experiments, we placed the ODE of an oscillator into the policy and learned its parameters. We simulated the ODE using the Euler method. While this ODE is not considered a neural ODE, this experiment could be easily extended to a setting in which some parameters of the ODE (for instance, a friction coefficient) are estimated by a neural network.
> >
> > > All Minor Remarks
> >
> > Thank you very much for pointing out all these inconsistencies. All of them are fixed in the latest version of the paper. For the causality dilemma, we added another sentence to make it more clear. We used Mio. as an abbreviation for million, which was not correct.
> >
> > We thank the reviewer again for the detailed review. We genuinely believe that his suggestions significantly improved the quality of our paper!

---

### Official Review · Reviewer_HaqJ · 2023-11-03

**Soundness:** 4 excellent
**Presentation:** 4 excellent
**Contribution:** 3 good
**Rating:** 8
**Confidence:** 4

**Summary:**

The paper proposes a novel method for decomposing a stateful policy into a stochastic internal state kernel and a stateless policy, resulting in a new policy gradient that is applicable to POMDPs without the need for backpropagating through time (BPTT). At the heart of this technique is the modification of the policy to output not only an action but also a prediction over the subsequent internal state. The authors have derived both stochastic and deterministic policy gradient theorems for this framework and have expanded the variance analysis of Papini et al. for policy gradient estimators. The experimental results demonstrate that the proposed method rivals algorithms that use full BPTT while requiring considerably less computational effort.

**Strengths:**

- The paper is very clear and provides a thorough presentation of the theory proposed by the method. I believe that this work will serve as a good reference for any work building on alternatives to BPTT for POMDPs.

- To my knowledge, the results presented are novel.

- The authors have conducted a detailed analysis of the algorithm across several complex tasks.

- While the theory occasionally presents straightforward extensions of classical policy gradient results, it is explained with exceptional clarity both in the main text and in the appendix.

**Weaknesses:**

- Many details, such as the results for the memory task, are relegated to the appendix. Nonetheless, I do not regard this as a significant weakness, given that the main text is already very dense with foundational results.

- Only ten seeds are utilized for the experiments, although it is well-known that MuJoCo tasks are prone to considerable variances in performance. I would recommend increasing the number of seeds to twenty for the final evaluation.

- The paper would benefit from including an analysis of the variance of the gradient for both the proposed method and BPTT, even if on a very simple benchmark. Additionally, it would be beneficial to examine the issues of vanishing and exploding gradients for both BPTT and the proposed method in at least one benchmark.

**Questions:**

- The occupancy measure is introduced without defining $z$

- Is the initial internal state learned, or is it initialized to zero at the beginning of each episode?

- Does the limit of the stochastic policy gradient converge to the deterministic policy gradient when the variance of the action and subsequent internal state approaches zero? In other words, is there a result analogous to Theorem 2 in the Deterministic Policy Gradient paper?

---

> ### Author Response · Authors · 2023-11-16
> **Answer to Reviewer HaqJ**
>
> We very much appreciate the detailed comments and suggestions of the reviewer. We have
> updated the main paper to address the concerns of the reviewer.
> We also added more detailed results, such as tables with explicit runtimes of all approaches in all
> environments in the appendix.
>
> We have highlighted all changes in the paper in
> blue to make it easier for the reviewer to spot the differences. In the following,
> we answer the reviewer's specific questions.
>
> >Many details, such as the results for the memory task, are relegated to the appendix. Nonetheless, I do not regard this as a significant weakness, given that the main text is already very dense with foundational results.
>
> We admit that the experiment section was not self-contained. We updated the paper now to give an overview of all experiments in a new Figure. Every experiment is now also getting a dedicated paragraph. Hence, the memory task itself and some of the results are now also shown in the main paper. In doing so, we reduced the amount of references to the appendix, making the experiment section easier to read.
>
> >Only ten seeds are utilized for the experiments, although it is well-known that MuJoCo tasks are prone to considerable variances in performance. I would recommend increasing the number of seeds to twenty for the final evaluation.
>
> We agree with the reviewer that 10 seeds might not be enough. We are currently running all experiments again to increase the number of seeds to 25 for all experiments in the final version of the paper.
>
> >The paper would benefit from including an analysis of the variance of the gradient for both the proposed method and BPTT, even if on a very simple benchmark. Additionally, it would be beneficial to examine the issues of vanishing and exploding gradients for both BPTT and the proposed method in at least one benchmark.
>
> This is indeed a very good idea. We are currently working on a toy example to *empirically* evaluate the variance of both gradient estimators in the policy gradient setting. We believe that this is a great addition to the theoretical analysis done in the paper.
>
> >The occupancy measure is introduced without defining z
>
> Thanks for pointing this out! This is fixed in the updated paper.
>
> >Is the initial internal state learned, or is it initialized to zero at the beginning of each episode?
>
> We always initialize the policy state to zero. We made this more clear in the updated paper. However, the initial state of the policy is a great way to induce any kind of knowledge a priori into the policy. For RNNs, we are not aware of a useful initial state, but in the case of an ODE, there are many interesting initial states you might want to try out.
>
> >Does the limit of the stochastic policy gradient converge to the deterministic policy gradient when the variance of the action and subsequent internal state approaches zero? In other words, is there a result analogous to Theorem 2 in the Deterministic Policy Gradient paper?
>
> This is a very good question. While we genuinely believe that a similar Theorem exists for our stateful policy gradient, this requires a proof. As the proof of Theorem 2 in the Deterministic Policy Gradient paper is non-trivial, we need time to carefully do the proof. We will try to do this for the final version of the paper.
>
> We thank the reviewer again for his suggestions and very much appreciate the time and effort spent.

---

> > ### Author Response · Authors · 2023-11-23
> > **Update on the toy example**
> >
> > Dear reviewer,
> > we want to inform you that we are currently finalizing the experiments regarding the toy example to show how our gradient approximator scales with (1) the dimensionality of the state (and action-space) and (2) the horizon. Our preliminary results confirm our insights from the other experiments, indicating that our gradient approximator scales effectively, with state-space dimensionality and horizon, but has higher variance in comparison to BPTT if the dimensionality is low and the horizon is short. We will add these experiments together with the updated 25 seeds experiments in the final version of the paper. Many thanks for proposing this idea.

---

### Official Review · Reviewer_gHcm · 2023-11-08

**Soundness:** 3 good
**Presentation:** 3 good
**Contribution:** 2 fair
**Rating:** 5
**Confidence:** 3

**Summary:**

The paper addresses the problem of learning policies with long term  history. Traditional methods often employ recurrent architectures, which rely on a persistent latent state that is modified at every time increment. However, this presents a challenge: when calculating the gradient of a loss function in such architectures, the gradient needs to be back-propagated through all preceding time steps. This process can lead to either vanishing or exploding gradients, making the training of recurrent models particularly difficult, especially as the historical data size increases. To address this issue, the authors introduce an alternative approach wherein the model's internal state is represented as a stochastic variable that is sampled at each time step. As a result, the state's stochastic nature prevents the direct computation of an analytical gradient, thereby circumventing the issues associated with backpropagation over time. The paper goes on to adapt established theoretical frameworks to this new model and suggests a method for incorporating actor-critic techniques. Empirical validation is conducted on a range of environments that are structured as Partially Observable Markov Decision Processes (POMDPs) by omitting certain observations.  It is shown that the proposed model achieves reasonable performance in comparison to BPTT-based approaches.

**Strengths:**

The paper is well-written and easy to follow. The supplementary appendix, which was not examined in detail, appears to be a valuable extension of the main text. The concept of characterizing the policy's internal state as a stochastic variable is intriguing and yields a sophisticated formulation. Additionally, the paper offers robust theoretical contributions and presents a methodology to modify conventional algorithms to encompass this concept.

**Weaknesses:**

I am not convinced by the arguments of the authors. In their formulation, even if using stochastic states prevents one from computing an analytic gradient, I don't understand why it would solve the problem of capturing long-term information. Indeed, when computing p(a_t,z_t|s_t,z_{t-1}), then this probability depends on the previous timestep, and so on, such that finding a good solution to the problem would need to propagate the loss to the previous timesteps to capture a good sequence of states. So there is still backpropagation through time, even if it is not made by the analytical gradient.

Then, usually, relying on stochastic variables decreases the sample efficiency of training methods. This is why people are using for example the reparametrization trick that allows one to compute an analytical gradient over a stochastic variable, to speed up training. Here the authors are claiming the opposite. So there is one point that I didn't catch in this paper, and I would like the authors to better explain why using stochastic nodes would avoid the problem of propagating information to the previous timesteps, and why they would expect a better sample efficiency than using an analytical gradient

Using stochastic variables as state of a policy is something made typically when using Thompson sampling-like methods. Papers like "Efficient Off-Policy Meta-Reinforcement Learning via Probabilistic Context Variables" are also using a stochastic internal state. How do you position your work w.r.t these approaches ?

In the experiments, it is not clear how the z distribution is modeled, and there is no discussion about possible choices and their possible impact. For instance, what about using a multinomial distribution? Discussing that point would be interesting.

Figure 1 is misleading since there are no arrows between the s nodes and the z nodes in the graph on the right and it seems that the sequence of z does not depend on the observations

**Questions:**

(see previous comments)

**Details Of Ethics Concerns:**

No concerns

---

> ### Author Response · Authors · 2023-11-15
> **Answer to Reviewer gHcm**
>
> We very much appreciate the detailed comments of the reviewer. We have updated the main paper to address the reviewer's concerns.
>
> Note that we additionally improved the experiments section by making the latter more self-contained and less reliant on the appendix. We also added more detailed results, such as tables with explicit runtimes of all approaches in all environments, in the appendix. We have highlighted all changes in the paper in blue to make it easier for the reviewer to spot the differences. In the following, we answer to the specific concerns of the reviewer.
>
> >I am not convinced by the arguments of the authors. In their formulation, even if using stochastic states prevents one from computing an analytic gradient, I don't understand why it would solve the problem of capturing long-term information. Indeed, when computing $p(a_t,z_t|s_t,z_{t-1})$, then this probability depends on the previous timestep, and so on, such that finding a good solution to the problem would need to propagate the loss to the previous timesteps to capture a good sequence of states. So there is still backpropagation through time, even if it is not made by the analytical gradient.
>
>
> In the BPTT algorithm, the gradients of the policy are computed analytically for the whole history up until time $t$. In contrast, our approach uses stochastic exploration to estimate the stateful policy gradient locally at time $t$ given an estimate of the cumulative return. This is basically the same principle as applying the likelihood-ratio trick used in standard reinforcement learning to estimate the gradient of a  policy in an unknown environment. In reinforcement learning, it is not possible to estimate the gradient of the policy without sufficient exploration. We follow the same principle to estimate the gradient of a stateful policy without relying on the analytical gradient of the history.
>
> While it is possible to interpret our approach as backpropagating information implicitly using our stochastic gradient estimator, we prefer the forward-looking perspective (the expected cumulative reward) used in reinforcement learning. This forward perspective becomes even more prominent when looking at our action value $V(s, z)$, i.e., the expected cumulative when being in the states $s$ and $z$, and sampling $a$ and $z'$  from our policy from there on. In other words, our policy needs to learn in the early states of a trajectory what information to encode for the future rather than propagating information back in time.
>
> It is also noteworthy that we show that the gradient of the policy can also be computed deterministically if we have access to a critic $Q(s, a, z, z')$. However, to estimate this critic, exploration is still necessary. With the conventional critic $Q(s, a)$, BPTT would still be necessary to propagate the information back in time.
>
> >Then, usually, relying on stochastic variables decreases the sample efficiency of training methods. This is why people are using for example the reparametrization trick that allows one to compute an analytical gradient over a stochastic variable, to speed up training. Here the authors are claiming the opposite. So there is one point that I didn't catch in this paper, and I would like the authors to better explain why using stochastic nodes would avoid the problem of propagating information to the previous timesteps, and why they would expect a better sample efficiency than using an analytical gradient
>
> We would like to point out that our approach is not more sample-efficient than BPTT. In fact, we agree with your statement that the efficiency in samples is generally lower compared to backpropagation through time, and this is something we see in the empirical results (robust RL results). What we claim is that our gradient estimator is more *time-efficient* than BPTT as the calculation of the gradient of a trajectory is an inherently sequential computation. This is particularly evident when taking the full history to compute the gradient. While BPTT can become more time-efficient by truncating the history, this leads to a biased gradient estimate. In contrast, our approach always provides an unbiased gradient estimate. Also, our gradient estimator is less dependent on the architecture of the policy, making it applicable to any stateful policy (c.f., theoretical analysis). For BPTT, specialized architectures like GRUs and LSTMs were introduced to cope with exploding and vanishing gradients. We see better efficiency in samples only when scaling the complexity of the tasks (Ant, and Humanoid in the robust RL, and all imitation learning experiments). We trace this back to the high dimensionality of the state space, which makes BPTT harder to optimize.

---

> > ### Author Response · Authors · 2023-11-15
> > **Further Answer to Reviewer gHcm**
> >
> > >Using stochastic variables as state of a policy is something made typically when using Thompson sampling-like methods. Papers like "Efficient Off-Policy Meta-Reinforcement Learning via Probabilistic Context Variables" are also using a stochastic internal state. How do you position your work w.r.t these approaches ?
> >
> > This is indeed a work that we were missing in the related work, but it is now added. While these methods look similar at first glance (especially due to similar notation), they do not constitute stochastic stateful policies. They rather span a distribution over a latent state given context -- in the case of the mentioned work, the history. Then, this latent variable distribution is explicitly trained to condense information about the history. The way these methods process the context (or the history) in the distribution over latent variables is through networks that either use a window of observations or BPTT. Hence, these methods neither use a stochastic stateful method nor approximate the gradient of a stateful policy stochastically.
> >
> > Also note that our gradient estimator allows us to train arbitrary stateful policy even beyond RNNs. We agree that this might not have been clear in the first version of the paper. Now, we provide an example in which we train the parameters of an ODE as a stateful policy in the main experiment section. This is especially interesting when an inductive bias is needed in a policy.
> >
> > >In the experiments, it is not clear how the z distribution is modeled, and there is no discussion about possible choices and their possible impact. For instance, what about using a multinomial distribution? Discussing that point would be interesting.
> >
> > We used Gaussians to model the distribution over $z'$. We made it more clear in the latest version of the paper. Our theoretical framework supports multinomial distribution and, in theory, any other distribution. The multinomial distribution corresponds to a discrete set of policy states. We believe that this could be a great idea for policies that need to remember discrete events. However, having a discrete policy state that is sampled at each step has non-trivial implications, which we would like to investigate in future work. Also, a discrete variable setting could not be implemented in BPTT due to the inherent non-differentiability. The lack of comparability to BPTT places this distribution class beyond the scope of this work. Due to the strong parallels of our gradient estimator to classical RL, the discussion about different continuous distributions for $z'$ is analogous to the one about the distributions over $a$ in classical RL, which is why we focussed on Gaussians. We believe that a full discussion about the intertwined effects between different distributions across $a$ and $z'$ deserves to be done in a dedicated work.
> >
> > >Figure 1 is misleading since there are no arrows between the s nodes and the z nodes in the graph on the right and it seems that the sequence of z does not depend on the observations
> >
> > The reviewer is correct. These paths were indeed missing and were added in the latest version of the paper.
> >
> > We thank the reviewer again for the time and effort spent. We hope that our answers, together with the updated paper, clarify the concerns. If some parts are still unclear or some changes in the paper are desired, we would be happy to answer and update the paper.

---

### Meta-Review · Area_Chair_vVrx · 2023-12-06

**Metareview:**

The reviewers agree that the paper presents an interesting alternative to BPTT for learning in POMDPs. Apart from fairly consistent questions on the presentation, clarity, there is still agreement that the paper provides interesting experiments and theoretical support for their method.

**Justification For Why Not Higher Score:**

consistent feedback across reviewers regarding the presentation of the paper

**Justification For Why Not Lower Score:**

the novelty and experimental/theoretical support justify acceptance.

---

### Decision · Program_Chairs · 2024-01-16

Accept (poster)